# Validation of Pan-Arctic Soil Temperatures in Modern Reanalysis and Data Assimilation Systems

Tyler C. Herrington [1], Christopher G. Fletcher [1], and Heather Kropp [2]

[1]Department of Geography and Environmental Management, University of Waterloo, 200 University Ave., Waterloo, Ontario, Canada, N2L 3G1
[2]Environmental Studies Program, Hamilton College, 198 College Hill Road, Clinton, 13323, New York, U.S.A.

**Correspondence:** Christopher G. Fletcher (chris.fletcher@uwaterloo.ca)

**Abstract.** Reanalysis products provide spatially homogeneous coverage for a variety of climate variables in regions such as the Arctic where observational data are limited. Soil temperatures are an important control of many land-atmosphere exchanges and hydrological processes, and permafrost soils, which contain roughly twice the amount of carbon in the atmosphere, are melting as the climate warms, releasing methane, a potent greenhouse gas. However, very little validation of reanalysis soil temperatures in the Arctic has been performed to date, because widespread in situ reference observations have historically been limited there. Here we validate pan-Arctic soil temperatures from eight reanalysis and land data assimilation system products, using a newly-assembled database of in situ observations from diverse measurement networks across Eurasia and North America. We examine product performance across the extratropical northern hemisphere between 1982 and 2018, and find that most products have soil temperatures that are biased cold by 1-5 K, with an RMSE of 2-9 K, and that biases and RMSE are generally largest in the cold season. Monthly mean values from most products correlate well with in situ data (r > 0.9) in the warm season, but show lower correlations (r = 0.55 - 0.85) in the cold season. Similarly, the magnitude of monthly variability in soil temperatures is well captured in summer, but overestimated by 20% to 50% for several products in winter. The suggestion is that soil temperatures in reanalysis products are subject to much higher uncertainty when the soil is frozen and/or when the ground is snow-covered, suggesting that the representation of processes controlling snow cover in reanalysis systems should be urgently studied. We also validate the ensemble mean of all available products, and find that, when all seasons and metrics are considered, the ensemble mean generally outperforms any individual product in terms of its correlation and variability, while maintaining relatively low biases. As such, we recommend the ensemble mean soil temperature product for a wide range of applications, such as the validation of soil temperatures in climate models, and to inform models that require soil temperature inputs, such as hydrological models.

## 1 Introduction

Soil temperatures, both near the surface, and at depth, are an important control of many physical, hydrological, and land surface processes, as soils act as a reservoir for energy and moisture underground. They provide an important initial condition for numerical weather prediction, as energy and water fluxes from the land are important for convective processes (Dirmeyer et al., 2006; Kim and Wang, 2007; Siqueira et al., 2009). As soils react relatively slowly to variations in weather, soil temper-

ature is also an important predictor of seasonal and mid-term weather forecasts (Xue et al., 2011). Soils over large portions of the Arctic are perennially frozen (permafrost soil). Roughly 1400 to 1600 gigatonnes of carbon (GtC) is estimated to be stored in soils in permafrost affected regions of the Northern Hemisphere (Hugelius et al., 2014). Continued warming, and thawing of permafrost soils, and related decomposition of carbon could act as a potential positive feedback on warming, by releasing more methane ($CH_4$) and carbon dioxide ($CO_2$) into the atmosphere (Koven et al., 2011).

In situ based soil temperature monitoring networks using thermistor probes, particularly at high latitudes, are limited in terms of their spatial and temporal coverage (Yi et al., 2019), making it difficult to assess hemispheric scale changes in permafrost. Reanalysis products have been used in a variety of weather and climate applications to provide information on a regular spatial grid; particularly in regions where limited or no observational data is available (Koster et al., 2004; Zhang et al., 2008). Previous studies validating reanalysis soil temperature have primarily focused on the middle latitudes, such as across China (Yang and Zhang, 2018; Zhan et al., 2020; Zhao et al., 2022), the Qinqhai-Tibetan Plateau (Hu et al., 2019; Jiao et al., 2023; Wu et al., 2018), Europe (Albergel et al., 2015; Johannsen et al., 2019), and the continental United States (Albergel et al., 2015; Xia et al., 2013), with a couple of recent studies validating soil temperatures globally (Li et al., 2020). Relative to in situ ground temperature probe networks, most reanalysis products are biased cold by about 2° C - 5° C, on average (Hu et al., 2019; Qin et al., 2020; Yang and Zhang, 2018). Ma et al. (2021) found that most reanalysis products show larger cold biases over polar regions than they do over tropical and temperature regions, while a recent study by Cao et al. (2020) found that ERA5-Land soil temperatures were biased warm over the Arctic, particularly in winter.

Several explanations have been suggested for the biases in reanalysis soil temperatures, including model parameterizations (Albergel et al., 2015; Cao et al., 2020; Chen et al., 2015; Wu et al., 2018; Xiao et al., 2013), air temperature biases (Cao et al., 2020; Hu et al., 2017), errors in topography and elevation, arising from the coarse resolution of reanalysis products (Yang and Zhang, 2018; Zhao et al., 2008; Ma et al., 2021), and errors in simulated snow cover and snow thermal insulation (Cao et al., 2020; Royer et al., 2021; Cao et al., 2022).

While soil temperature biases in individual reanalysis products may limit their utility, a consensus is emerging that multi-product ensembles, based on the same principle as ensemble weather prediction (World Meteorological Organization, 2012), are an effective way to increase the signal-to-noise ratio for many important geophysical variables. Ensemble mean datasets based on combinations of in situ, model, satellite and reanalysis data have been used to reduce biases in estimates of snow water equivalent (Mudryk et al., 2015), soil moisture (Dorigo et al., 2017; Gruber et al., 2019), precipitation (Beck et al., 2019), as well as for local scale permafrost simulations (Cao et al., 2019). Li et al. (2020) suggest that a similar method could be used to reduce biases in reanalysis soil temperatures.

Reanalysis soil temperatures have been relatively well characterized over the middle latitudes. Studies validating Arctic soil temperatures in reanalysis products, however, have either focused on a singular product (Cao et al., 2020), or have only considered a limited spatial extent (Li et al., 2020; Ma et al., 2021).

Here we perform a validation of pan-Arctic (and Boreal) soil temperatures from eight reanalysis and land data assimilation system (LDAS) products. The main objectives are to 1) validate the 8 reanalysis and LDAS soil temperature products in terms

of their bias, RMSE, correlation and standard deviation, and 2) investigate whether an ensemble mean soil temperature product outperforms the individual reanalysis products.

## 2 Data

### 2.1 Reanalysis and LDAS Data

Table 1 and 2 outline the six reanalysis and two LDAS soil temperature products used in this study. For simplicity, the term "reanalysis" will hereafter be used to describe both reanalysis and LDAS products. A summary of each product follows below. Products were remapped onto the European Reanalysis - Interim (ERA-Interim) grid for comparison, using three different methods: nearest neighbour, bilinear interpolation, and first-order conservative remapping. The choice of remapping method did not affect the overall conclusions of the study, and the analysis is based on data remapped using the conservative remapping method, as it facilitated the use of the largest number of validation sites and grid cells.

The reanalysis products investigated span a wide range of horizontal resolutions, ranging between $0.1°$, in the case of ERA5-Land and FLDAS, to $0.75°$ for ERA-Interim (Table 1). Most products - CFSR, ERA-Interim, ERA5, ERA5-Land, and the Famine Early Warning Systems Network Land Data Assimilation System (FLDAS) simulate soil temperature across 4 vertical layers, while MERRA2 includes 6 vertical layers, and JRA-55 calculates soil temperature across a single layer. The topmost soil layer has the highest resolution (7 cm to 10 cm in most cases), while the bottom soil layer often averages soil properties over a metre or more (Table 2).

The Noah Land Surface Model (Noah-LSM) (Chen et al., 1996; Betts et al., 1997; Koren et al., 1999; Ek, 2003) is used by CFSR and FLDAS. CFSR uses the Noah-LSM in a fully coupled mode to obtain a first-guess land-atmosphere simulation, before operating in a semi-coupled mode with GLDAS to obtain information about the state of the land surface (Saha et al., 2010). FLDAS, however, is run in an offline mode, utilizing meteorological forcing from MERRA2 (McNally et al., 2017), and rainfall information from NOAA's Global Data Assimilation (GDAS) (Derber et al., 1991), the Climate Hazards group Infrared Precipitation with Stations (CHIRPS) (Funk et al., 2015), and the African Rainfall Estimation version 2.0 (RFE2) (Xie and Arkin, 1997).

ERA-Interim, ERA5 and ERA5-Land use versions of the Tiled ECMWF Scheme for Surface Exchanges over Land (TES-SEL) land model (Viterbo, 1995; Viterbo and Betts, 1999). In the case of ERA-Interim, TESSEL is informed by empirical corrections from 2m (surface) air temperature and humidity (Dee et al., 2011). Meanwhile, ERA5 and ERA5-Land use an updated version of TESSEL, known as the Hydrology-Tiled ECMWF Scheme for Surface Exchanges over Land (HTESSEL) (Balsamo et al., 2009). In ERA5, a weak coupling exists between the land surface and atmosphere. It includes an advanced LDAS that incorporates information regarding the near-surface air temperature, relative humidity, as well as snow cover (de Rosnay et al., 2014), along with satellite estimates of soil moisture and soil temperature from the top 1 metre of soil (de Rosnay et al., 2013). ERA5-Land, unlike ERA5, does not directly assimilate observational data. Instead, the ERA5 meteorology (such as air temperature, humidity and atmospheric pressure) is used as forcing information for HTESSEL; allowing it to be run at higher resolutions (Muñoz-Sabater et al., 2021). It includes an improved parameterization of soil thermal conductivity allow-

**Table 1.** Summary of the 8 reanalysis and LDAS products, their equatorial resolution, land model, and relevant references.

| Product | Data Period | Resolution | Land Model | References |
|---|---|---|---|---|
| CFSR | 1979 - 2010 | 0.31° x 0.31° | Noah LSM | Saha et al. (2010) |
| CFSv2 | 2011 - Present | 0.2° x 0.2° | Noah LSM | Saha et al. (2014) |
| ERA5 | 1940 - Present | 0.25° x 0.25° | HTESSEL | Hersbach et al. (2020) |
| ERA5-Land | 1950 - Present | 0.1° x 0.1° | HTESSEL | Muñoz-Sabater et al. (2021) |
| ERA-Interim | 1979 - Aug 2019 | 0.75° x 0.75° | TESSEL | Dee et al. (2011) |
| FLDAS | 1982 - Present | 0.1° x 0.1° | Noah LSM | McNally et al. (2017) |
| JRA55 | 1956 - Present | 0.56° x 0.56° | Simple Biosphere Model | Harada et al. (2016) |
| | | | | Kobayashi et al. (2015) |
| MERRA2 | 1980 - Present | 0.5° x 0.625° | Catchment LSM | Gelaro et al. (2017) |

**Table 2.** Summary of the 8 reanalysis and LDAS products and the number and depths of the soil layers included. *The JRA-55 Simple Biosphere Model contains up to three soil layers (whose depths vary depending on vegetation type), but the soil temperature is averaged over all layers to produce a singular value at each grid cell.

| Product | Soil Layers | Soil Depths (in cm) |
|---|---|---|
| CFSR | 4 | 0 - 10, 10 - 40, 40 - 100, 100 - 200 |
| CFSv2 | 4 | 0 - 10, 10 - 40, 40 - 100, 100 - 200 |
| ERA5 | 4 | 0 - 7, 7 - 28, 28 - 100, 100 - 289 |
| ERA5-Land | 4 | 0 - 7, 7 - 28, 28 - 100, 100 - 289 |
| ERA-Interim | 4 | 0 - 7, 7 - 28, 28 - 100, 100 - 289 |
| FLDAS | 4 | 0 - 10, 10 - 40, 40 - 100, 100 - 200 |
| JRA55 | 3* | temperature averaged over soil column |
| MERRA2 | 6 | 0 - 9.88, 9.88 - 29.4, 29.4 - 67.99, |
| | | 67.99 - 144.25, 144.25 - 294.96, 294.96 - 1294.96 |

ing for it to account for ice content in frozen soil, improvements to soil water balance conservation, and the ability to capture rain-on-snow events (Muñoz-Sabater et al., 2021).

MERRA2 utilizes the the Catchment Land Surface Model (CLSM) (Ducharne et al., 2000; Koster et al., 2000). Though MERRA2 does not include a land surface analysis (Gelaro et al., 2017), CLSM is informed using an updated version of the Climate Prediction Center unified gauge-based analysis of global daily precipitation (CPCU) precipitation correction algorithm that originated in MERRA-Land (Reichle et al., 2017b). No corrections are available, however, for high latitude regions north of 62.5° N (Reichle et al., 2017a). Finally, JRA-55 uses the Simple Biosphere Model (SiB) (Onogi et al., 2007; Sato et al., 1988; Sellers et al., 1986) in an offline mode, forced by atmospheric data and data from land surface analyses that incorporate microwave satellite retrievals of snow cover (Kobayashi et al., 2015).

## 2.2 Observational Data

Owing to the lack of dense soil temperature monitoring networks in the Arctic, most of the observed soil temperature record is characterized by a soil temperature record that is temporally and spatially sparse (Luo et al., 2020). While Russia has a more complete record of permafrost temperatures extending back to the 1980s (Sherstiukov, 2012), longer term term permafrost records over North America are generally limited to the western Arctic Smith et al. (2010). Portions of coastal Nunavik, in Northern Quebec, have had permafrost records since the 1990s CEN (2020a, b, c, d, e, f, g), while soil temperature measurements in the Central Arctic are rather sparse Smith et al. (2010). Rather than limit our validation to a small geographic region in the permafrost zone, as several prior studies have done (Hu et al., 2019; Qin et al., 2020; Wu et al., 2018; Ma et al., 2021; Li et al., 2020), we choose to combine data from a variety of sparse and dense networks. Such an approach has been used to validate soil temperature and permafrost performance in ERA5-Land (Cao et al., 2020), and allows for the examination of larger geographic regions, as well as for the inclusion of a more diverse set of vegetation types across the continent (Ma et al., 2021).

The study compiles a comprehensive set of in situ soil temperature measurements, approximately 1700 stations in total, from across extratropical Eurasia and North America (Table 3, and Supplemental Metadata). Incorporating data from multiple diverse sparse networks, the dataset includes data from the Yukon Geological Survey (Yukon Geological Survey, 2021), the Northwest Territories (Cameron et al., 2019; Ensom et al., 2020; Gruber et al., 2019; Spence and Hedstrom, 2018a, b; Street et al., 2018), Roshydromet Network in Russia (Sherstiukov, 2012), Nordicana series D (Nordicana) (Allard et al., 2020; CEN, 2020a, b, c, d, e, f, g), Global Terrestrial Network for Permafrost (GTN-P) (GTN-P, 2018), and Kropp et al. (2020) - in an attempt to provide a representative estimate of soil temperature across the circumpolar Arctic. Our validation data also includes sites from outside regions typically underlain by permafrost, in order to facilitate a comparison of the performance of reanalysis soil temperatures at high latitudes with their performance in regions outside the permafrost zone. These include stations from Kropp et al. (2020), Sherstiukov (2012), and GTN-P (2018), as well as locations from the Manitoba Mesonet network (RoTimi Ojo and Manaigre, 2021), the Michigan Enviro-weather Network (MAWN) (Enviro-weather, 2022), the North Dakota Mesonet Network (NDAWN) (North Dakota Mesonet Network, 2022), and the Alberta Climate Information Service (ACIS) network (Alberta Agriculture, Forestry and Rural Economic Development, 2022). Data is also sourced from a peatland ecosystem in Metro Vancouver (Lee et al., 2017), and several locations in central and Northern BC (Déry, 2017; Hernández-Henríquez et al., 2018; Morris et al., 2021). This provides a unique baseline upon which to perform a hemispheric wide assessment of soil temperature in reanalysis and LDAS systems, and to the authors' knowledge, presents the most comprehensive analysis to date of soil temperatures across Canada and the Great Lakes basin.

## 2.3 Collocation of Station and reanalysis Data

In order to compare with data from reanalysis and LDAS products, temperatures were averaged across two depth bins: a near surface layer (0 cm to 30 cm), and soil temperatures at depth (30 cm to 300 cm). For each site, temperatures from all depths residing within a layer were averaged, producing an estimated layer averaged temperature for every time-step. In order

**Table 3.** Summary of the observational data networks included in the study, including the dataset name, number of stations included, and their references. Note that Nordicana D references are listed by site in the supplemental metadata.

| Dataset | Number of Sites | Reference |
|---|---|---|
| GTN-P | 68 | (GTN-P, 2018) |
| Heather Kropp | 229 | (Kropp et al., 2020) |
| Roshydromet Network | 458 | (Sherstiukov, 2012) |
| Nordicana D | 34 | See supplemental station metadata |
| NWT Open Report 2017-009 | 73 | (Cameron et al., 2019) |
| NWT Open Report 2018-009 | 214 | (Gruber et al., 2019) |
| NWT Open Report 2019-004 | 9 | (Ensom et al., 2020) |
| NWT Open Report 2019-017 | 31 | (Rudy et al., 2020) |
| Street and Wookey (2016) | 5 | (Street and Wookey, 2016) |
| Yukon Permafrost Database | 112 | (Yukon Geological Survey, 2021) |
| Baker Creek | 6 | (Spence and Hedstrom, 2018b) |
| Cariboo Alpine MesoNET | 12 | (Hernández-Henríquez et al., 2018) |
| | | (Déry, 2017) |
| | | (Morris et al., 2021) |
| Burns Bog | 1 | (Lee et al., 2017) |
| Manitoba Mesoscale Network | 85 | (RoTimi Ojo and Manaigre, 2021) |
| Enviro-Weather Network | 75 | (Enviro-weather, 2022) |
| ACIS | 31 | (Alberta Agriculture, Forestry and Rural Economic Development, 2022) |
| NDAWN | 150 | (North Dakota Mesonet Network, 2022) |

to maximize the amount of observational data available, layer-averaged soil temperatures were calculated at each timestep with
135 all available data. This tradeoff meant that layer averages often included a different number of depths at different timesteps, and as such, we needed to limit our analysis of soil temperature trends and variability to locations where layer averages had a consistent number of depths.

Many of the in situ (station) sites reported measurements at hourly or daily frequency, however we chose to perform the analysis at monthly time scales, in order to focus on processes controlling the seasonal cycle of soil temperatures. As such, we
use monthly averages of soil temperatures for validation purposes. Outlier observations with anomalies greater than $\pm\ 3.5\sigma$ were removed before monthly averaging.

Since the station data often included days with missing observations, the sensitivity of the monthly averages to missing data was tested, by computing monthly averages in five ways: using all months with at least one valid day in a month, using all months with at least 25, 50, and 75 percent valid data, and finally using all months with no missing data in a month. It was
145 found that T$_{soil}$ was not substantially impacted by the inclusion or exclusion of months containing missing data. In order to increase sample size, we therefore included all months with at least 50 percent valid data.

In order to be considered as a validation location, the grid cell was required to include soil temperature data for all eight reanalysis/LDAS products, and be collocated with at least one in situ station. Duplicate stations across datasets were excluded. In situ locations were only included if there was at least 2 years worth of in situ data, in order to properly assess the station's

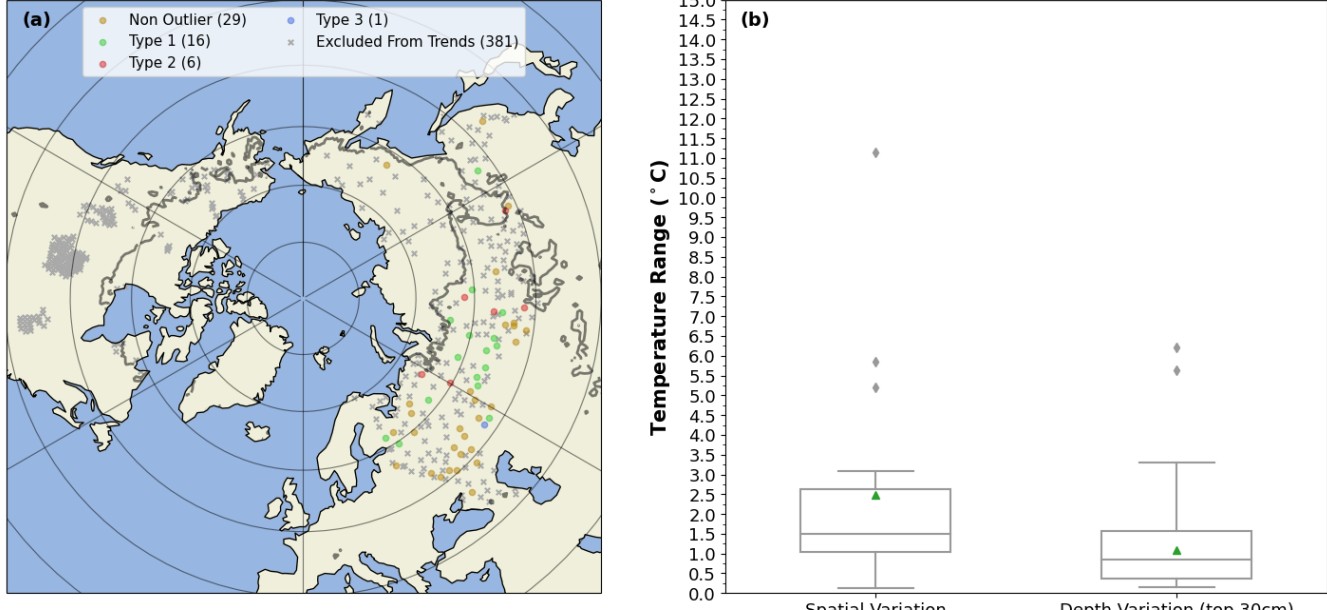

**Figure 1.** (a): location of the validation grid cells collocated with in situ stations in the near-surface layer. Grid cells excluded from the soil temperature trends analysis are shown as an "x". Type 1 refers to grid cells that where the ensemble mean simulates a winter minimum soil temperature that is too cold. Type 2 refers to grid cells where the ensemble mean simulates a summer maximum soil temperature that is too cold. Type 3 refers to grid cells where the ensemble mean underestimates the seasonal cycle of soil temperatures. The number in brackets beside each legend entry displays the number of grid cells in each category. The contour line encircles regions where the Obu et al. (2018) permafrost cover is at least 50 percent. (b): Impact of spatial variation and depth variation on the spread of soil temperatures in a grid cell. The mean is shown by a green triangle, and outliers are shown as grey diamonds.

seasonal cycle. For grid cells containing multiple in situ stations, the value used in the comparison is a simple spatial average of the in situ stations in that grid cell on each calendar day.

Over Eurasia, grid cells contained a single in situ measurement location. In North America, however, a number of the grid cells contain two or more in situ stations. The near surface layer layer includes 430 validation grid cells (Figure 1, panel A), while at depth, there are 377 grid cells (not shown). A subset of stations with longer timeseries and a more complete data

record are used to calculate soil temperature trends (Section 4.2). Stations included in the soil temperature trend and variability analysis are shown as circles of varying size and colour, while those excluded from the soil temperature trend and variability analysis are shown as an x (Figure 1, Panel A). The details of Figure 1 - Panel A will be described further in Section 4.2.

To calculate spatial averages, a simple average of (layer-averaged) soil temperatures from all stations within the bounds of a particular grid cell was calculated at each timestep, using all available stations. This meant that the number of stations included

at each timestep wasn't always consistent, and the analysis of soil temperature trends was limited to a subset of 52 grid cells in Eurasia where the following conditions were met:

1. The timeseries included data between Jan 1985 and Dec 2010, with no missing data.

2. The number of stations included in the spatially averaged grid cell temperature was consistent over all timesteps.

3. The number of depths included in the layer averaged soil temperature of each contributing station remained consistent over all timesteps.

As a result North American grid cells were excluded from the soil temperature trends analysis, and the analysis is based on grid cells from Eurasia (where grid cells often only contained a single station) (Figure 1, Panel A). Using a subset of grid cells that incorporate multiple stations in the spatial average, and include a consistent number of stations and depths in the timeseries, we quantify the variability in soil temperatures between stations within a grid cell, and across depths within a layer average. It was found that the median temperature range between stations within a grid cell was approximately 1.5° C, roughly 1.75 times larger than the median temperature range across depths within the near surface layer of a station (Figure 1, Panel B), suggesting that temperature variability within a grid cell is substantially larger than variations in temperatures within the near surface layer of a particular station.

## 3 Methods

### 3.1 Validation Metrics

Reanalysis/LDAS and observational (station) soil temperature data were collocated with one another spatially and temporally. Grid-cell level soil temperatures from each product were compared against in situ soil temperatures using the following statistical metrics: bias (Eq. 1), root-mean-squared-error (RMSE) (Eq. 2), normalized standard deviation ($\sigma_{norm}$) (Eq. 3 and Eq. 4), and the Pearson correlation (R) (Eq. 5). We also include an overall skill score for each model; a Thackeray et al. (2015) type formulation of the Taylor (2001) skill score (SS) (Eq. 6). Statistical metrics were calculated as follows:

$$Bias = \frac{1}{N} \sum_{n=1}^{N} (T_p - T_i) \tag{1}$$

$$RMSE = \sqrt{\frac{1}{N} \sum_{n=1}^{N} (T_p - T_i)^2} \tag{2}$$

$$\sigma = \sqrt{\frac{\sum_{n=1}^{N} (x_n - \overline{x})^2}{N - 1}} \tag{3}$$

$$\sigma_{norm} = \frac{\sigma_{T_p}}{\sigma_{T_i}} \tag{4}$$

$$R = \frac{\frac{1}{N}\sum_{n=1}^{N}(T_p - \overline{T_p})(T_i - \overline{T_i})}{\sigma_{T_p}\sigma_{T_i}} \tag{5}$$

$$SS = \frac{2(1+R)}{(\sigma_{norm} + \frac{1}{\sigma_{norm}})^2} \tag{6}$$

Where $T_p$ is the $\mathrm{T}_{soil}$ from the reanalysis product, and $T_i$ is the $\mathrm{T}_{soil}$ of the in situ data. $\overline{T_p}$ and $\overline{T_i}$ refer to the mean $\mathrm{T}_{soil}$ of the reanalysis product and in situ data, respectively, while $N$ is the number of monthly soil temperature values. $\sigma_{norm}$ refers to the normalized standard deviation, while $\sigma_{T_p}$ and $\sigma_{T_i}$ are the standard deviations of the reanalysis product soil temperatures and in situ soil temperatures, respectively. Finally, $x$ refers to the $\mathrm{T}_{soil}$ (from a particular timestep in a dataset), $\bar{x}$ is the mean $\mathrm{T}_{soil}$ of the dataset, $R$ is the Pearson correlation, and *SS* refers to the skill score.

Metrics were calculated separately for each individual grid cell, and then averaged to obtain regional values. Estimates for the permafrost zone and the zone with little to no permafrost were also calculated by averaging together metrics from grid cells falling within a particular zone. Skill scores were calculated separately for the near surface, and depth, while the "overall" skill score represents an average of the near surface and depth skill scores.

## 3.2 Binning of Datasets by Season and Permafrost

Datasets were binned into a cold season and warm season using the University of East Anglia's Climatic Research Unit (CRU) TS version 4.07 2m air temperature ($\mathrm{T}_{air}$) (Harris et al., 2020) for each grid cell. Cold season months are those where $\mathrm{T}_{air} \leq$ -2° C, while the warm season refers to months with $\mathrm{T}_{air} >$ -2° C, where $\mathrm{T}_{air}$ is the monthly mean air temperature. Sensitivity testing on the cold/warm season revealed no substantive impact on our conclusions using a threshold of 0° C, -5° C, and -10° C. We also tested the impact of using a different temperature dataset to perform the binning; the ERA5 2m air temperature, which resulted in similar findings.

Permafrost zonation was estimated using the Obu et al. (2018) permafrost map, which employs a temperature at the top of the permafrost (TTOP) model based on a 2000-2016 climatology, driven by a combination of remotely sensed land surface temperatures, downscaled atmospheric data from ERA-Interim, and landcover information from The European Space Agency (ESA) Climate Change Initiative (CCI) (Obu et al., 2019). To maximize the sample size in each group, we merge the 'continuous' and 'discontinuous' permafrost zones into a single category called the 'permafrost zone', and compare against the zone with 'little to no permafrost', which includes all regions with <50% permafrost cover.

**Table 4.** Number of grid cells in each elevation bin for the near surface and at depth.

| Elevation Range | Near Surface Grid Cells | Depth Grid Cells |
|---|---|---|
| Below 500 m | 310 | 275 |
| 500 to 1000 m | 105 | 87 |
| 1000+ m | 15 | 15 |

### 3.3 Elevation Impacts

The authors examined the potential impacts of elevation differences between in situ datasets and reanalysis products by
215 estimating the station elevation using the 90 m Copernicus Global Digital Elevation Model (GLO-90) (European Space Agency,
2021) and obtaining reanalysis elevations at their native resolution for the nearest grid cell to the station. For grid cells with
more than one station, station elevations were averaged together to obtain a grid cell estimate of the "station" elevation.

Grid cells were grouped into three elevation bins based on the station elevation, and it was found that over 70% of grid
cells are located in regions where the in situ station(s) are below 500 m (Table 4). Only 15 grid cells had station elevations
above 1000 m, so the authors grouped all grid cells at or above 500 m together for the purposes of validation. While reanalysis
products generally underestimated the elevation of higher elevation station, with an average RMSE of between 144 m and 589
m (not shown), this did not appear to have a major impact on soil temperature performance. Readers are referred to Section 4.3
(Spatial Variability) for more details.

### 3.4 Regridding of Reanalysis Products and Calculation of Ensemble Mean Soil Temperature

The ensemble mean soil temperature product is a "blended" soil temperature product based on a simple average of soil
temperatures from six individual soil temperature products (CFSR, ERA-Interim, ERA5, ERA5-Land, FLDAS and MERRA2).
Owing to JRA55's simplified land model, which is unable to capture near surface soil temperatures, we decided to exclude it
from the ensemble mean product, as its inclusion dramatically increased the bias and RMSE of the ensemble mean. Two soil
temperature estimates: one of the "near-surface", and another of soil temperatures at depth are calculated for each timestep. The
230 near-surface soil temperature is based on the average soil temperature of the top 2 soil layers from each product – representing
an estimate of the average soil temperature in the top 30cm. The "deep" soil temperature estimate is based on an average of
the soil temperatures from layers further down the soil column, down to a maximum depth of about 300cm. While the vertical
discretization is coarser than that of the individual products, this approach allows the ensemble mean product to incorporate
soil temperatures from products with different land models, whose vertical resolution is not constant.

All products were first re-gridded to the ERA-Interim grid using a first-order conservative remapping technique (Jones,
1999). The near surface soil layers were calculated as a simple average of the top 2 soil layers in each reanalysis product (except
for JRA-55 which only includes a single soil layer that represents the temperature averaged over the entire soil column). Soil
temperatures at depth were calculated as a simple average of all layers whose bottom depth is within the top 300 cm of soil. For
CFSR/CFSv2, ERA-Interim, ERA5 and ERA5-Land, this represented the third and fourth soil layers, while the third, fourth

and fifth soil layers were included from MERRA2. For JRA55, we were again limited to the single averaged soil layer. Readers are referred to Table 2 for further information about product soil layers.

After the near surface and deep soil layer average temperatures were calculated for each product, the ensemble mean soil temperature, for each layer, was calculated as the unweighted arithmetic mean of the eight products for each month and for each grid cell.

## 4    Validation of reanalysis and LDAS Products

### 4.1    Extratropical Northern Hemisphere Mean

Most products show annual mean skill scores (purple) ranging between 0.9 and 0.96. In general, skill scores are higher near the surface where soil temperatures are more correlated with air temperatures (Figure 2, Table S1 and S2). JRA55 is a noticeable outlier (skill score = 0.54), as it uses a simplified land model where the soil temperatures are averaged across the

250 soil column. Thus its soil temperatures underestimate the seasonal cycle of observed soil temperatures in the near surface, and the timing of annual maximum and minimum soil temperatures is offset by roughly a month (as deep soil temperatures are slower to react to changes in air temperature or surface energy balance changes) (not shown).

For the most part, reanalyses show small to moderate negative (cold) biases in both seasons, though ERA5-Land exhibits a small positive (warm) bias in winter (Figure 2, Table S1 and S2). JRA55 exhibits larger biases with respect to near surface

soil temperatures, as it underestimates the annual range of near surface temperatures, whereas, at depth, biases are smaller, and the skill score is higher, reflective of the fact that its soil temperatures are more reflective of deeper soil layers. Generally speaking, most products show a maximum cold (negative) bias when soil temperatures are between -2° C to -10° C, and there is a tendency for the biases of most products to decrease or flip sign over the coldest temperatures. There is also a larger spread in bias over the coldest range of temperatures (Figure 3). In the case of JRA55, however, the maximum cold bias occurs

over the warmest temperatures, and the bias flips sign when soil temperatures are near zero, becoming warm (positive) for station soil temperatures below 0° C (Figure 3), resulting from JRA55's reduced seasonal cycle of soil temperatures. With ERA-Interim, the largest cold (negative) biases are found over the coldest temperatures - likely linked to issues with its snow cover representation (discussed in Section 6.1).

For individual products, the variability in reanalysis soil temperature for a given observed soil temperature (as measured by

265 their standard deviation) is generally greatest over frozen soil conditions (particularly temperatures below -20° C) - evidence of the reduced agreement between product soil temperatures and observations. In addition, there is a larger range of temperatures displayed for a given observed soil temperature in the cold season than in the warm season (not shown). The spread in standard deviation between products (similar to their biases), is also generally largest over colder temperatures. The reduced standard deviation near the surface in the -32° C bin is likely a function of the small sample size (11). JRA55 is an exception, as it shows

a maximum standard deviation when soil temperatures are near freezing, and variance decreases thereafter (Figure 4) - likely due to the fact that it underestimates the magnitude of the coldest temperatures (not shown).

## 4.2 Seasonal Cycle

Strong seasonal differences exist in reanalysis performance - particularly near the surface, where skill scores are 0.08 to 0.35 lower during the cold season than in the warm season, and there is a noticeably larger spread (greater disagreement) between products. The skill at depth shows less seasonal variation, but is still noticeably lower during the cold season in most cases, with most products showing a decline in skill of between 0.02 and 0.08. The decline in cold season skill is mirrored by increases in near surface bias and RMSE for several products - particularly ERA-Interim, whose bias and RMSE are 4.1° C, and 3.7° C larger, respectively. Interestingly, biases for all products are somewhat larger in the warm season at depth, though seasonal differences are also generally smaller in the deeper soil layers (Figure 2).

JRA55 shows a 4.2° C positive (warm) bias during the cold season, and a 7.0° C negative (cold) bias in the warm season 2) - suggesting that the seasonal cycle in soil temperatures is too small. Meanwhile, ERA5-Land displays a small warm (positive) bias during the cold season; a feature not present in the warm season. This is suggestive that snow cover properties may be driving the winter warm bias in ERA5-Land (which will be discussed further in Section 6.1).

Similar seasonal variation is present in reanalysis soil temperature correlations (against station data), as most products show warm season correlations of greater than 0.93 near the surface 5). Meanwhile the near surface cold season correlations are generally lower by approximately 0.16 to 0.39 (Figure 5) - which contributes to lower skill scores (Figure 2). The poor JRA55 correlation near the surface arises from its mismatched seasonal cycle.

Most products generally capture the observed soil temperature variance during the warm season, as normalized standard deviations are within 25% of the observed for all products. This is contrasted by the cold season, where several products overestimate soil temperature variability (particularly at depth), contributing to a decline in product skill. Moreover, there is a larger spread in variance during the cold season - suggesting that there is less agreement between the products themselves (Figure 5). ERA5-Land's (blue diamond) cold season skill is impacted by its underestimation of cold season soil temperature variability (which is roughly half of the near surface observed variance), and arises in part because of its warm (positive) bias in winter (Figure 2). ERA-Interim (lime-green square) shows unrealistically large soil temperature variability over the cold season over both depths, while CFSR (purple), JRA55 (red) and FLDAS (black) are too variable at depth (Figure 5). This contributes to a substantial decline in their cold season skill (Figure 2).

## 4.3 Spatial Variability

Soil temperature performance over the permafrost zone is typically worse relative to the performance over the zone with little to no permafrost. Near Surface skill scores are generally reduced by 0.05 - 0.1 over the annual mean, and by as much as 0.17 for ERA-Interim, while at depth, annual mean skill scores are reduced as much as 0.26 (Figure S1). The RMSE for most products is typically 1.3° C to 4.5° C larger over the cold season, and larger by 0.1° C to 2.1° C in the warm season over the permafrost zone (Figure S1). Meanwhile, the spread in standard deviation between products, at depth, is roughly 1.6 to 3.7 times larger over the permafrost zone, relative to the zone with little to no permafrost (Figure S1), because of substantial differences in the variance of ERA5-Land, JRA55 and ERA-Interim. It remains for future studies to determine whether these

differences are due to the regions being colder, or due to structural issues with the land models, though this is beyond the scope of this study. The differences in correlation and standard deviation between the permafrost zone, and the zone with little to no permafrost, in the near surface soil layers are less dramatic (Figure S2).

The ERA5-Land warm (positive) bias in the cold season is largest over permafrost regions (Figure 6) - particularly over Siberia and across North America (not shown). In the case of JRA55, however, the warm biases over the cold season are largest further south. In fact, over many grid cells in the permafrost zone, JRA55 exhibits a cold (negative) bias during the cold season (not shown).

Generally speaking, the skill is higher over Eurasia than over North America (Figure 7). The lower skill in North America arises in part due to the underestimation of seasonal cycle over many grid cells in the Yukon, and an overestimation of variability of cold season temperatures over much of the Great Lakes Region (Figure S3). CFSR and JRA55 are an exception, however, as they greatly overestimate the cold season variability over much of western Eurasia (not shown), and consequently exhibit lower Eurasian skill scores. Soil temperature correlations (with in situ soil temperatures) are also lower by about 0.02 to 0.08 for most products in the warm season over North America, relative to Eurasia (not shown), which further contributes to reduced skill over North America.

As there are few stations above 1000m, elevation does not have a substantial impact on product performance (Figure 8). The slight improvement in near surface cold season performance in CFSR at higher elevations can be linked to a slightly higher correlation, while in ERA-Interim, the slightly higher skill score is due to a slight improvement in cold season temperature variance (not shown). Skill scores at depth are lower by about 0.05 to 0.1 over higher elevation stations (not shown), mainly due to small reductions in correlation (not shown), however the overall conclusions are not altered.

## 4.4 Multi-Annual Trends

We calculate product trends over the 1985-2010 period in order to be able to calculate a station estimate from a subset of 55 Eurasian grid cells that have a continuous timeseries, and a consistent number of sites and depths included over all dates and times (denoted as the Eurasian subset from hereon in). Trends at depth are very similar in magnitude and spatial pattern to the near surface, so we focus on the near surface results here. Trends in the Eurasian subset (hatched lines) are generally representative of the Eurasian average (blue), though are overestimated slightly in the case of CFSR.

Regionally averaged 1985-2010 annual mean soil temperature trends show a small positive trend of <0.5° C in most products, over both Eurasia and North America, and trends in most products are generally consistent with the station estimate over the Eurasian subset, with the exception of CFSR (Figure 9, Panel C). In CFSR (purple), the trend is near zero over North America, and tends towards negative in Eurasia. This arises because of anomalously cold years in 2009 and 2010, and anomalously warm periods in the 80s and early 90s at the beginning of the timeseries (particularly over Eurasia) (Figure 9, Panels A and B). It is likely that the cold anomalies in 2009 and 2010 can be linked to issues with CFSR snow cover between January 2009 and January 2011 (Figure S4).

Similar to skill score, and RMSE, products show greater disagreement over higher latitudes (Figure S5), and during winter (Figure S6). ERA5 in particular, and ERA5-Land , and ERA-Interim to a lesser extent, show several pockets of cooling over

**Table 5.** Standard deviation (as a measure of spread between products) of the mean biases in winter minimum and summer maximum soil temperature, as a function of latitude and depth (from Figures 10 and S8, Panels C and D). Latitude bands are 10 degrees in width, such that the 40° N latitude band is an average between 40° N and 50° N, while the 60° N latitude band is an average between 60° N and 70° N, for example.

| Latitude Band | Near Surface | | Depth | |
|---|---|---|---|---|
| | Winter Minimum | Summer Maximum | Winter Minimum | Summer Maximum |
| 40°N | 2.80° C | 2.32° C | 1.51° C | 1.25° C |
| 50°N | 2.80° C | 2.41° C | 1.56° C | 1.43° C |
| 60°N | 3.81° C | 2.95° C | 2.41° C | 1.90° C |

Siberia and the western Arctic over North America (Figure S5), driven by strong cooling trends in DJF (Figure S6). While FLDAS, JRA55 and MERRA2 show a pocket of cooling in DJF over western North America, they do not exhibit the same degree of cooling over Siberia as CFSR or the European reanalyses, and in some cases even exhibit warming (Figure S5). Trends over JJA show good agreement between products, with most regions showing small warming trends of < 1° C, and pockets of slight cooling over portions of Eurasia and Western North America (Figure S7). Several products also show a pocket of stronger warming, of around 1.5 to 2° C in SE Eurasia (Figure S7).

### 4.5 Variability in Seasonal Extremes

As discussed in earlier sections, the mean soil temperatures in most products are generally biased cold (negative) in both the warm and cold seasons. For most products, this also extends to their winter minimum (Figure 10, Panel C) and summer maximum temperatures 10, Panel D), exhibiting a cold bias over all latitude bands.

While the spread between products remains relatively consistent across latitudinal bands over summer, the spread between products increases at higher latitudes over winter (Table 5). Using the standard deviation as a measure of spread between product biases, the standard deviation in winter minimum bias increases from 2.80° C over the 40° N latitude band, to 3.81° C north of 60° N. This is in large part to substantially colder biases in ERA-Interim (green) at higher latitudes. Meanwhile the standard deviation in the mean summer maximum bias sees smaller increases (from 2.32° C at 40° N, to 2.95° C at 60° N) (Table 5).

Similarly, at depth, winter minimum soil temperature sees a larger spread at high latitudes (increasing from a standard deviation of 1.51° C at 40° N to 2.41° C at 60° N), while the spread in the summer maximum sees less variation with latitude (Table 5).

Winter warm (positive) biases in ERA5-Land (sky-blue) are most prevalent over higher latitudes (Figure 10, Panel C). Interestingly, ERA-Interim (green) shows similar biases to ERA5 (cyan) and ERA5-Land (sky-blue) in summer and is one of the best performing products. This is suggestive that ERA-Interim's degraded performance over winter could be related to snow cover. The conclusions regarding variability in soil temperature extremes, at depth, are generally similar to those near the surface (Figure S8, Panel C and D), though the spread between products is not quite as large as it is near the surface (Table 5).

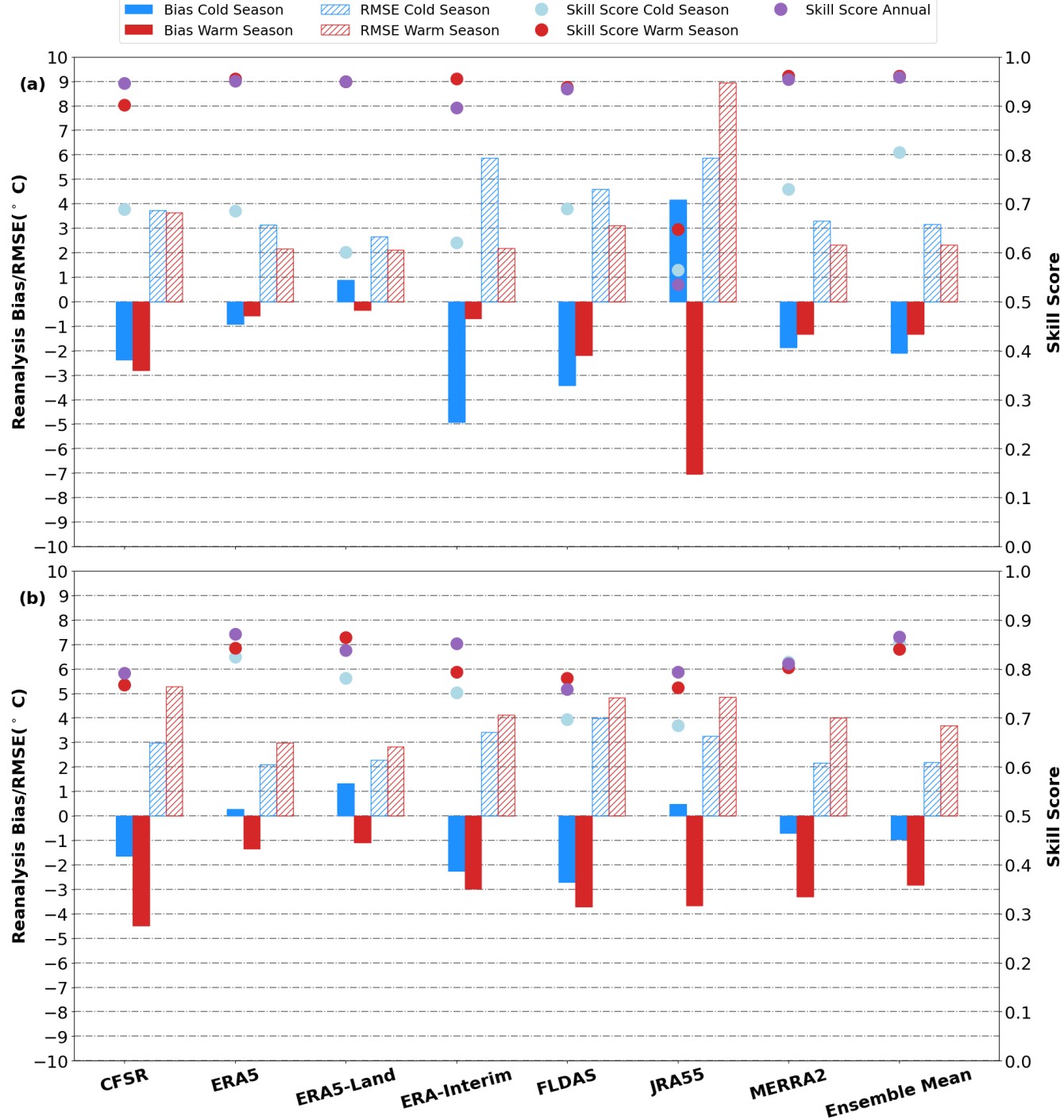

**Figure 2.** Bias (solid color), RMSE (hatching) and skill scores (circles) of each product the cold season (blue) ($\leq$ -2° C) and the warm season (red) (> -2° C) performance of reanalysis products. The skill score is also shown over the annual cycle (purple). (a) displays the bias, RMSE and skill score for the near surface (0 cm to 30 cm) layer, while (b) displays the bias, RMSE and skill score at depth (30 cm to 300 cm). The ensemble mean is shown beside for comparison.

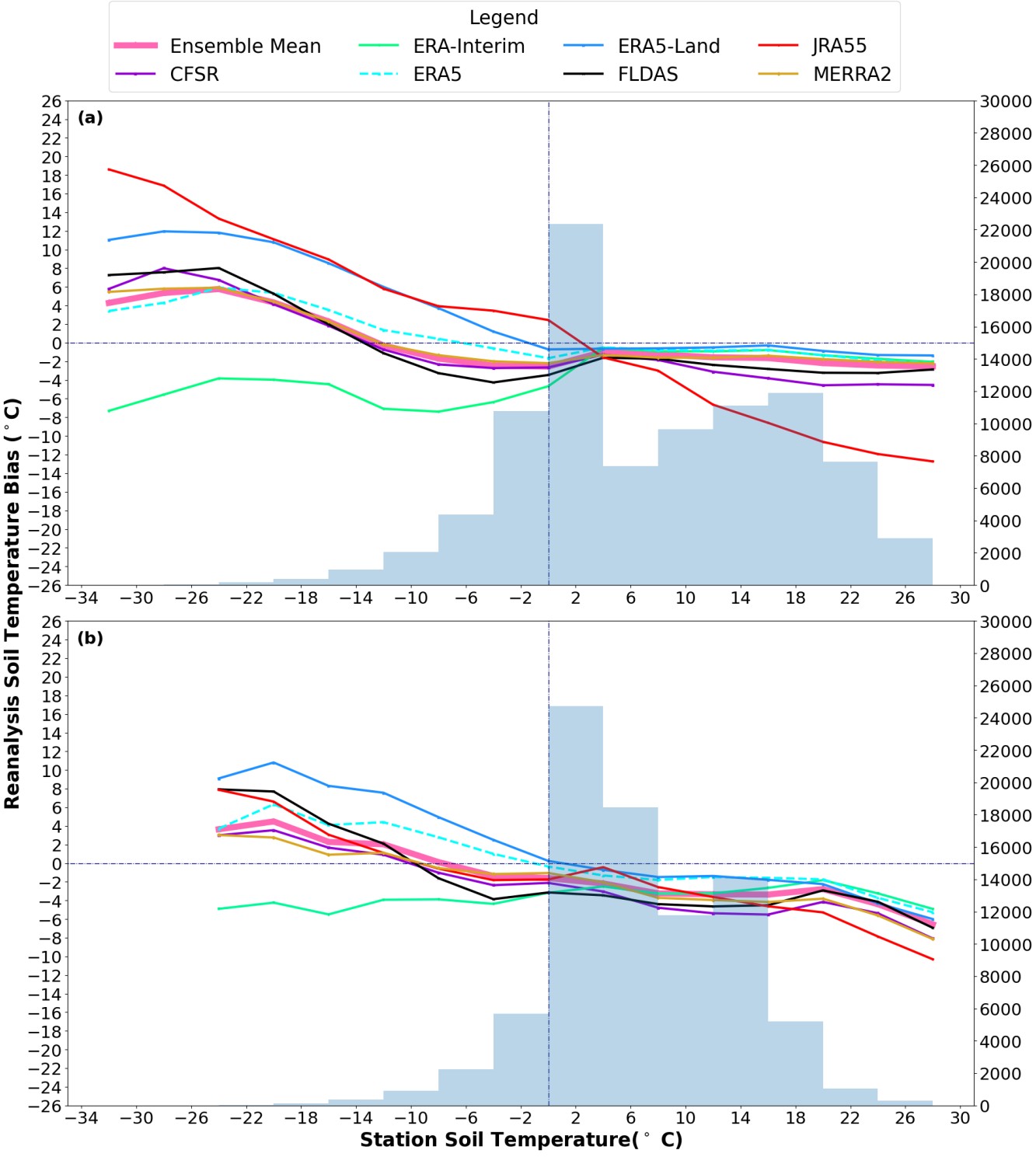

**Figure 3.** Reanalysis soil temperature bias as a function of station soil temperature for (a) the near surface (0 cm to 30 cm) layer, and (b) at depth (30 cm to 300 cm). Station temperatures are binned into 4° C intervals, beginning with the -32° C to -28° C bin, and ending with the 26° C to 30° C bin. The midpoint of each temperature bin is plotted along the x-axis. The secondary y-axis displays the number of datapoints in each bin (in conjunction with the histogram).

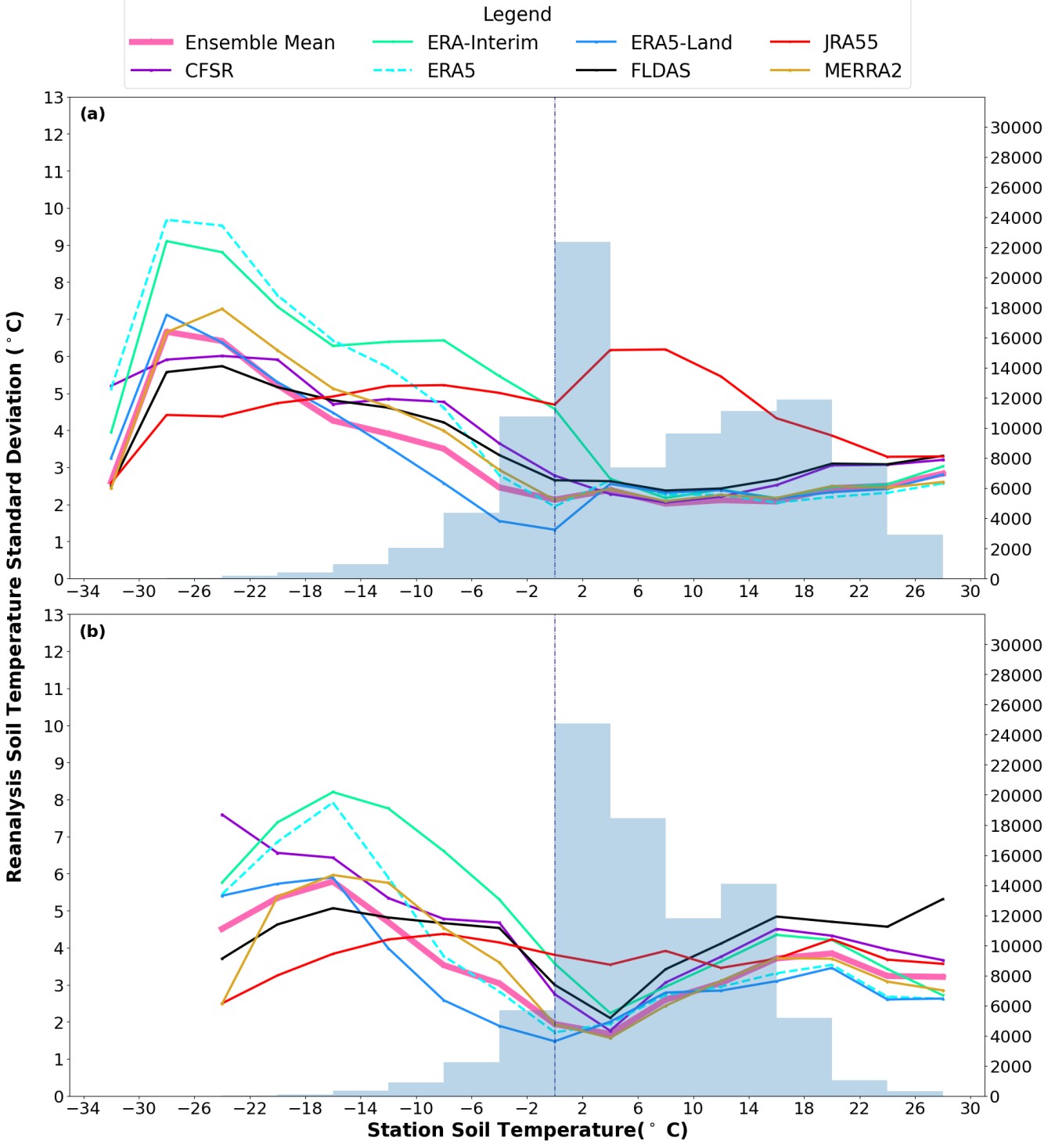

**Figure 4.** Reanalysis soil temperature standard deviation as a function of station soil temperature for (a) the near surface (0 cm to 30 cm) layer, and (b) at depth (30 cm to 300 cm). Station temperatures are binned into 4° C intervals, beginning with the -32° C to -28° C bin, and ending with the 26° C to 30° C bin. The midpoint of each temperature bin is plotted along the x-axis. The secondary y-axis displays the number of datapoints in each bin (in conjunction with the histogram).

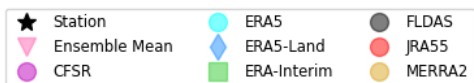

**Figure 5.** Taylor Diagram of the cold season and the warm season performance of reanalysis products. Panels A and B refer to the cold season, while panels C and D refer to the warm season. The top panels, (a) and (c) are for the near surface while the bottom panels, (b) and (d) refer to soil temperatures at depth. The concentric rings (solid grey lines) refer to the centralized root mean square error (CRMSE), and a product would have a CRMSE of zero if the timeseries of the reanalysis matched the station data perfectly; with a normalized standard deviation of one, and a correlation of one.

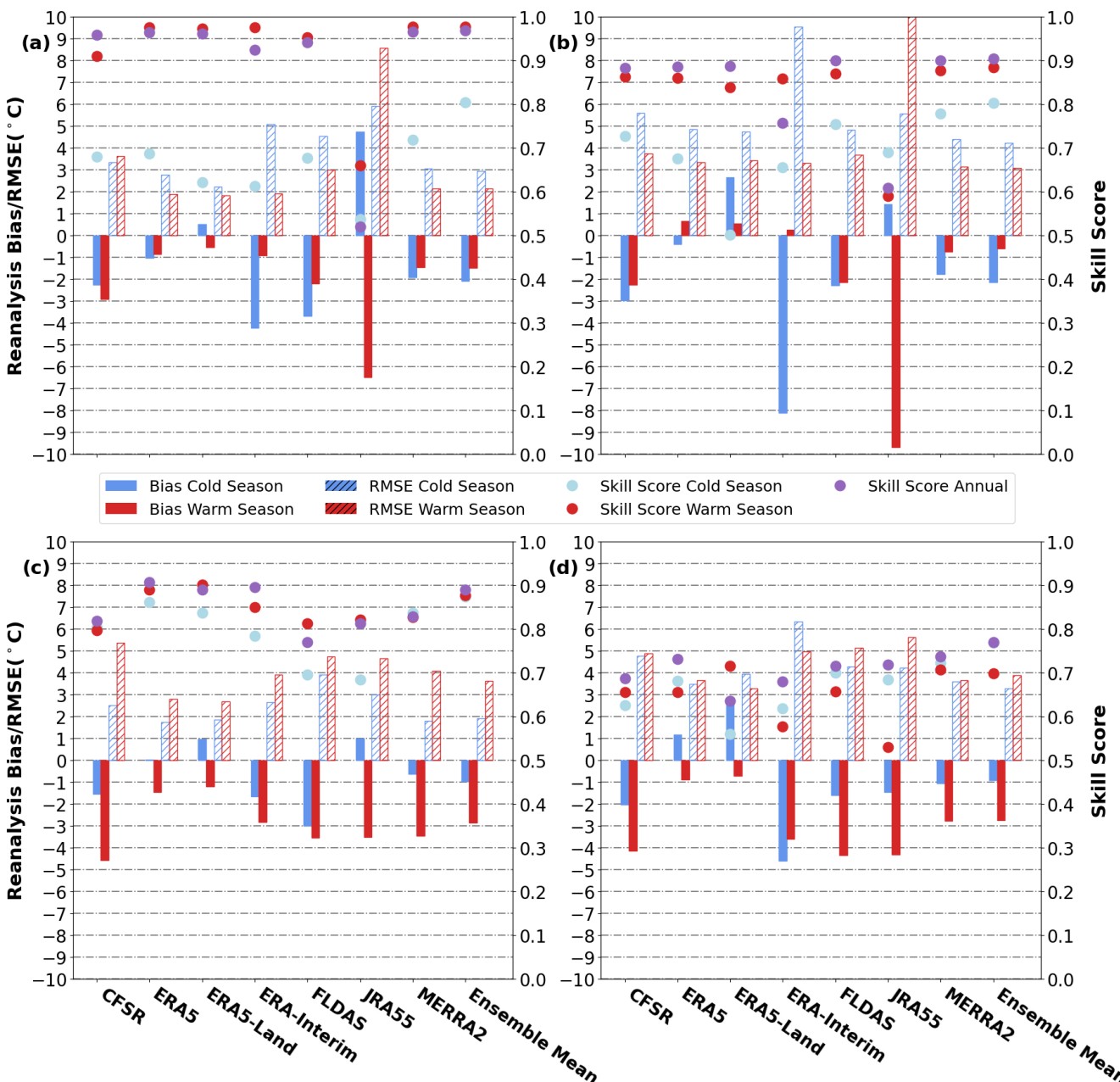

**Figure 6.** Bias (solid color), RMSE (hatching) and skill scores (circles) of each product the cold season (blue) and the warm season (red) performance of reanalysis products over the zone with little to no permafrost, (a) and (c), and the permafrost zone, (b) and (d). The skill score is also shown for the annual cycle (purple). (a) and (b) displays the bias, RMSE and skill score for the near surface layer, while (c) and (d) display the bias, RMSE and skill score at depth. The ensemble mean is shown beside for comparison.

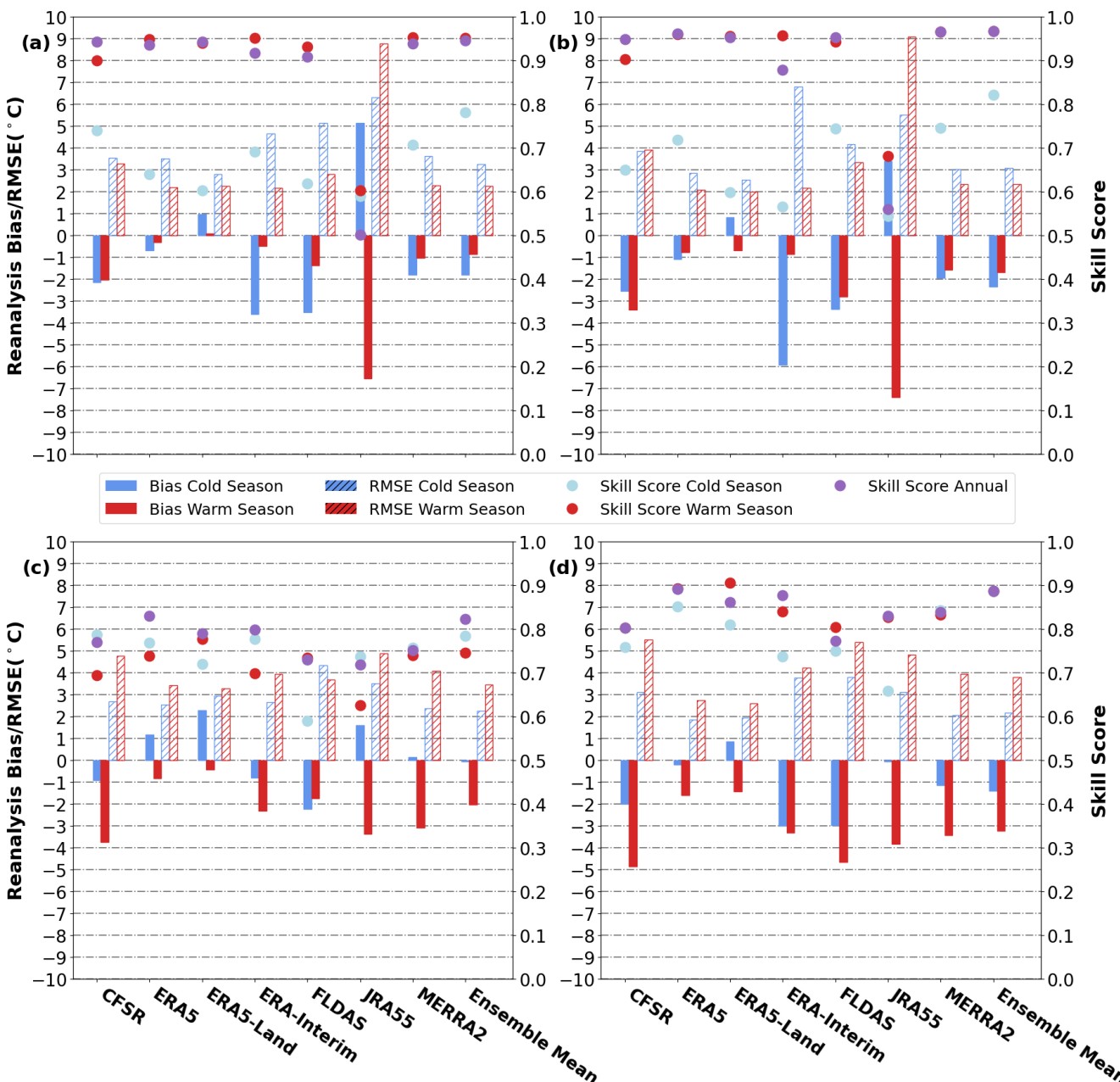

**Figure 7.** Bias (solid color), RMSE (hatching) and skill scores (circles) of each product the cold season (blue) and the warm season (red) performance of reanalysis products over North America, (a) and (c), and Eurasia, (b) and (d). The skill score is also shown for the annual cycle (purple). (a) and (b) displays the bias, RMSE and skill score for the near surface (0 cm to 30 cm) layer, while (c) and (d) display the bias, RMSE and skill score at depth (30 cm to 300 cm). The ensemble mean is shown beside for comparison

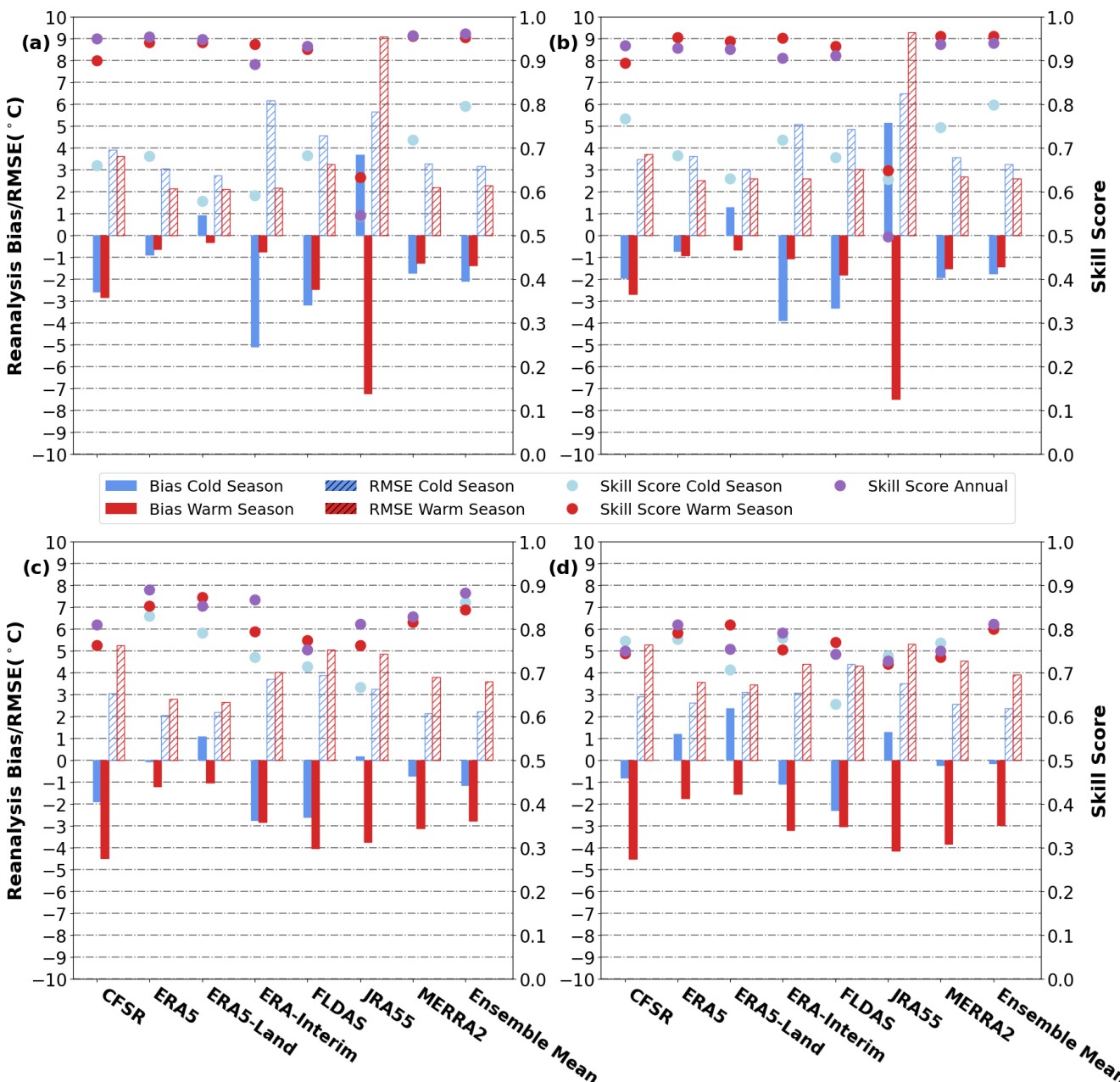

**Figure 8.** Bias (solid color), RMSE (hatching) and skill scores (circles) of each product the cold season (blue) and the warm season (red) performance of reanalysis products over low elevation grid cells (below 500m), (a) and (c), and grid cells at or above 500m elevation, (b) and (d). The skill score is also shown for the annual cycle (purple). (a) and (b) displays the bias, RMSE and skill score for the near surface (0 cm to 30 cm) layer, while (c) and (d) display the bias, RMSE and skill score at depth (30 cm to 300 cm). The ensemble mean is shown beside for comparison.

**Figure 9.** Near surface soil temperature anomalies and trends for each of the reanalysis products. (a) displays the regionally averaged 1982-2018 annual mean soil temperature anomalies for each reanalysis product north of 40° N over Eurasia, while (b) displays the same, but over North America. (c) exhibits an estimate of the regionally averaged 1985-2010 annual mean decadal soil temperature trend for each of the individual products, and the ensemble mean for comparison (the error bars represent the 95% CI for the mean trend).

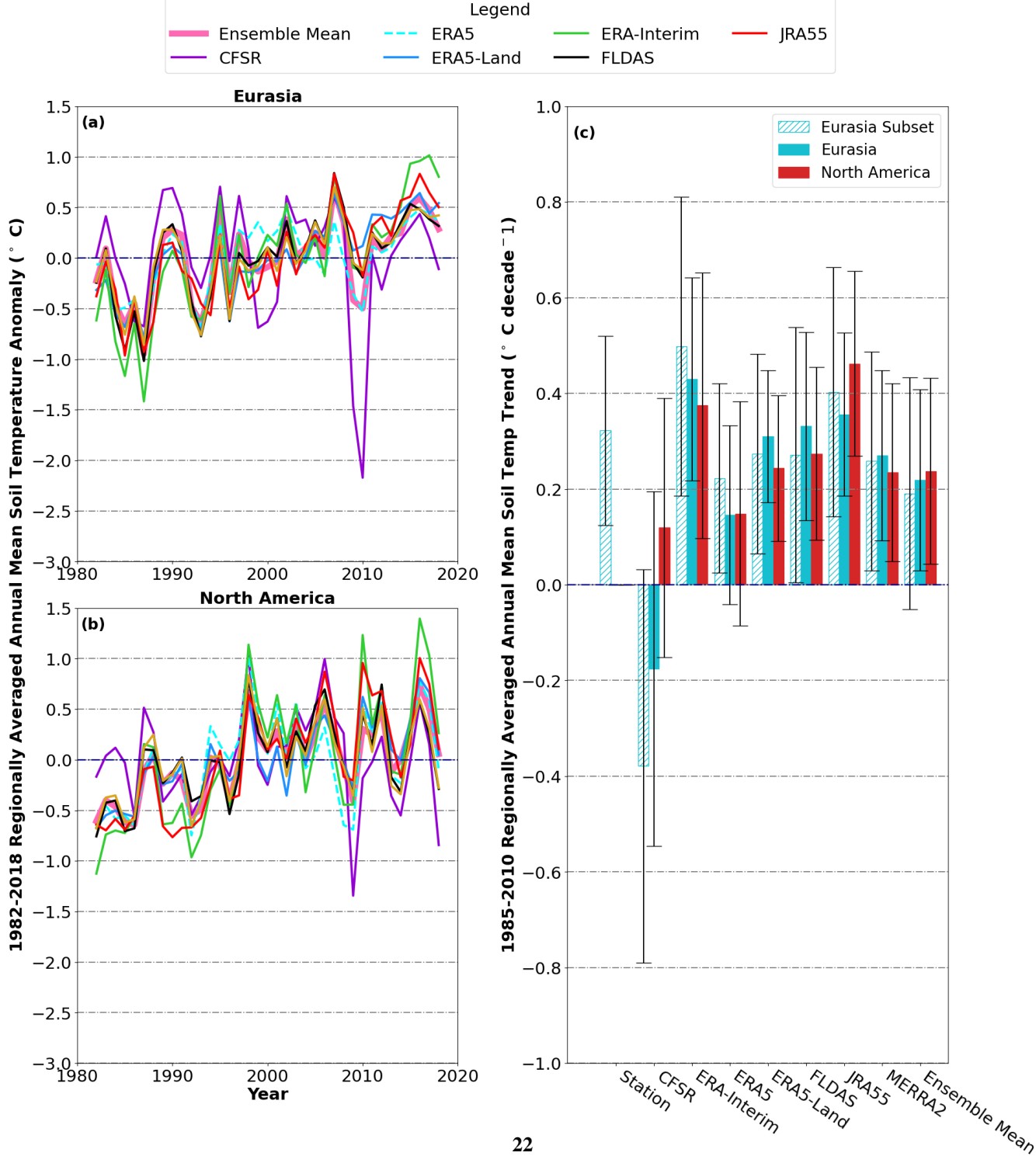

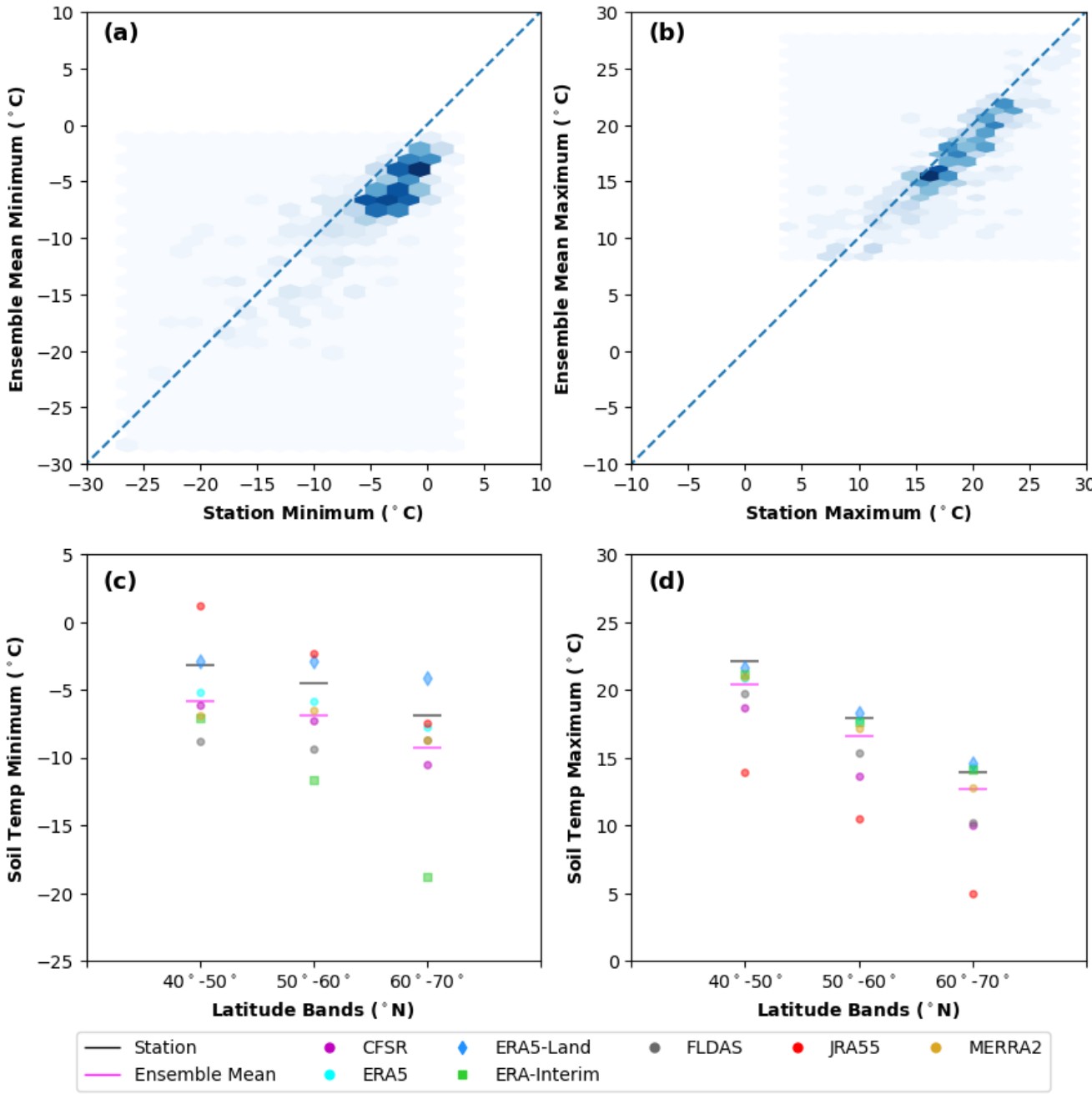

**Figure 10.** Performance of the near surface soil temperature variability in the Ensemble Mean. (a): Scatterplot of the station and ensemble mean winter minimum soil temperature. (b): Scatterplot of the station and ensemble mean summer maximum soil temperature. (c): latitudinal averages (from Eurasian grid cells) of near surface soil temperature winter minimum for the ensemble mean and contributing products. (d): latitudinal averages (from Eurasian grid cells) of near surface soil temperature summer maximum for the ensemble mean and contributing products.

## 5 Ensemble Mean Product

### 5.1 Validation

The ensemble mean soil temperature product produces a soil temperature dataset that shows closer agreement with observed soil temperatures than most/all individual products. The annual mean, ensemble mean skill score is higher than any individual product over the extratropical northern hemisphere (Figure 2). The bias of the ensemble mean soil temperature generally quite close in magnitude to the best performing products over both seasons depths (Figure 2). Moreover, the ensemble mean product (pink) displays a temporal variance within 20% of the observed variability over all depths. We find that the inclusion of a

greater number of products in the ensemble mean yields greater reductions in bias and RMSE, and analogous improvements in correlation, though the incremental improvement in skill begins to saturate beyond four products (not shown).

    The value of using the ensemble mean soil temperature is particularly noticeable in the cold season when individual products see a decline in skill, and a larger spread in performance. Bias and RMSE only see small increases over permafrost regions (Figure S1 and Figure 11), while products such as ERA5, and ERA5-Land, which have a small RMSE over more southern

regions, see more substantial increases in bias and RMSE over the permafrost zone (Figure S1)

    The near surface skill of the ensemble mean in the cold season is nearly 10% than the next best product (Figure 2). While several products fail to capture the cold season temperature variance, the variance of the ensemble mean product remains within 20% of the observed variability (Figure 5). In addition, the extratropical northern hemisphere mean cold season biases are close in magnitude to best performing product (Figure 2) over both depths, and its correlations are generally larger, by roughly 0.05,

than the best performing product over both depths (Figure 5). Thus, the ensemble mean soil temperature dataset provides the best estimate of in situ temperatures for the broadest range of conditions.

### 5.2 Ensemble Mean Multi-Annual Trends

    We focus our analysis of trends on the near surface data, as the spatial pattern of soil temperature trends near the surface and at depth show a pattern correlation of greater than 0.95 (not shown), and the conclusions regarding performance are generally

similar. Regionally averaged over the extratropical northern hemisphere north of $40°$ N, the ensemble mean shows a small positive trend of $0.23 \pm 0.09$ ° C decade$^{-1}$ over Eurasia, and $0.20 \pm 0.10$ ° C decade$^{-1}$ over North America (Figure 9, Panel C) between 1982 and 2018. Most regions see small positive annual mean soil temperature trends of $\leq 0.5°$ C decade$^{-1}$, though portions of Western North America and Siberia exhibit slight cooling trends of $< 0.5°$ C decade$^{-1}$ (Figure 12, Panel A).

    Annual mean soil temperature trends in the Ensemble Mean over Eurasia show a strong correlation of 0.82 with observations

(Figure 12 Panel B). The ensemble mean generally captures the spread in observed soil temperature trends (Figure 9, Panel C), and generally captures the correct sign of the trends across the 52 Eurasian stations (Figure 12, Panel B), though it has a tendency to slightly underestimate the trends (Figure 9, Panel C and Figure 12, Panel B). The spatial pattern in annual mean soil temperature trends at depth is nearly identical, displaying a pattern correlation of 0.98 (not shown). Similar to the near surface, most grid cells show a small positive soil temperature trend for the ensemble mean at depth, and a moderately strong

correlation with in situ soil temperature trends of 0.68 (not shown).

## 5.3 Ensemble Mean Variability in Seasonal Extremes

Similar to most products, the ensemble mean is biased cold in both the winter minimum (Figure 10, Panel A) and the summer maximum (Figure 10, Panel B) soil temperature. As the previous paragraph, and Figure 10, Panels A and B show, however, there is a fair degree of variability in the behaviour of the ensemble mean seasonal extremes – making an assessment of the mean behaviour in seasonal extremes somewhat tricky. Therefore, in the following paragraphs, we will focus on the most robust findings.

Near the surface, biases in the winter minimum soil temperature (Figure 10, Panel A) are generally larger than in the summer maximum (Figure 10, Panel B). This can also be seen in the latitudinally averaged soil temperatures, where the ensemble mean (pink line) is further from the station (black line) in the winter minimum (Figure 10, Panel C) than in summer maximum (Figure 10, Panel D), which agrees with the findings that the near surface cold season bias is generally larger than the bias in warm season (Figure 2).

At Depth, the findings differ a bit. Figure S8 Panel B shows that there is a noticeably greater disagreement between the summer maximum ensemble mean and in situ soil temperatures; especially over colder regions. It is also apparent that the latitudinally averaged biases in the summer maximum temperature (Figure S8, Panel D) are larger than their winter minimum counterparts (Figure S8, Panel C) - consistent with the findings that the extratropical mean bias in the warm season is larger than the bias in the cold season (Figure 2).

Referring to Figure 1, Panel A, several different types of grid cells are denoted. The first group - Type 1 (16 occurrences) grid cells are characterized by a strong cold bias in (underestimate) the winter minimum soil temperature (Figure 13, Panel A). A second grouping of grid cells, which we refer to as Type 2 (6 occurrences) grid cells are defined as those which have a strong warm bias in (overestimate) the summer maximum temperature (Figure 13, Panel B). A common feature of the third group, Type 3 (1 occurrence), is that they underestimate the observed seasonal cycle of soil temperatures (Figure 13, Panel C). Often the in situ station(s) located within Type 3 grid cells are located in areas devoid of vegetation, and it is likely that disagreements in the simulated vegetation cover in the contributing reanalysis products may partially account for the reduced seasonal cycle. Many grid cells in the Yukon, whose comprising stations are similarly located in areas devoid of vegetation, would also meet the criteria for a Type 3 outlier, as the ensemble mean normalized standard deviation (a measure of soil temperature variability) is substantially smaller than 1.0 in both seasons (Figure S3), though these grid cells were excluded as their timeseries were too short. If we examine all grid cells over North America, a fourth group can be identified: instances where the ensemble mean simulates a seasonal cycle of soil temperatures that is too large. This is evident in Figure S3, where a cluster of grid cells in the Great Lakes region show a normalized standard deviation much larger than 1.0, and for a number of grid cells in western Eurasia that were excluded because they didn't meet the strict requirements for trend analysis.

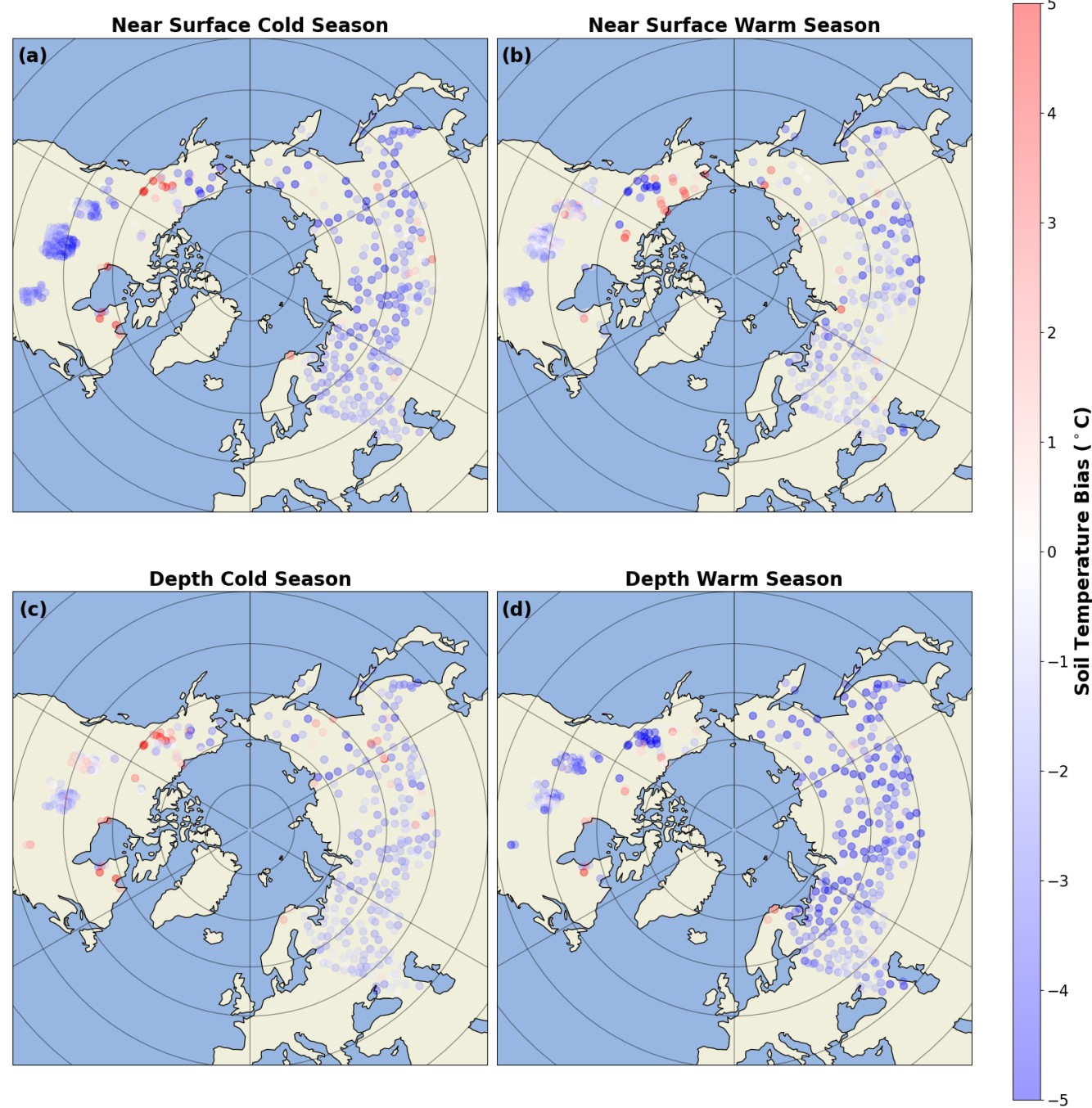

**Figure 11.** Spatial map of bias for the Ensemble Mean product. Values for the cold season are shown in the left hand panels, and those for the warm season are shown in the right hand panels. Panels A and B show the near surface bias, while biases at depth are shown in Panels C and D.

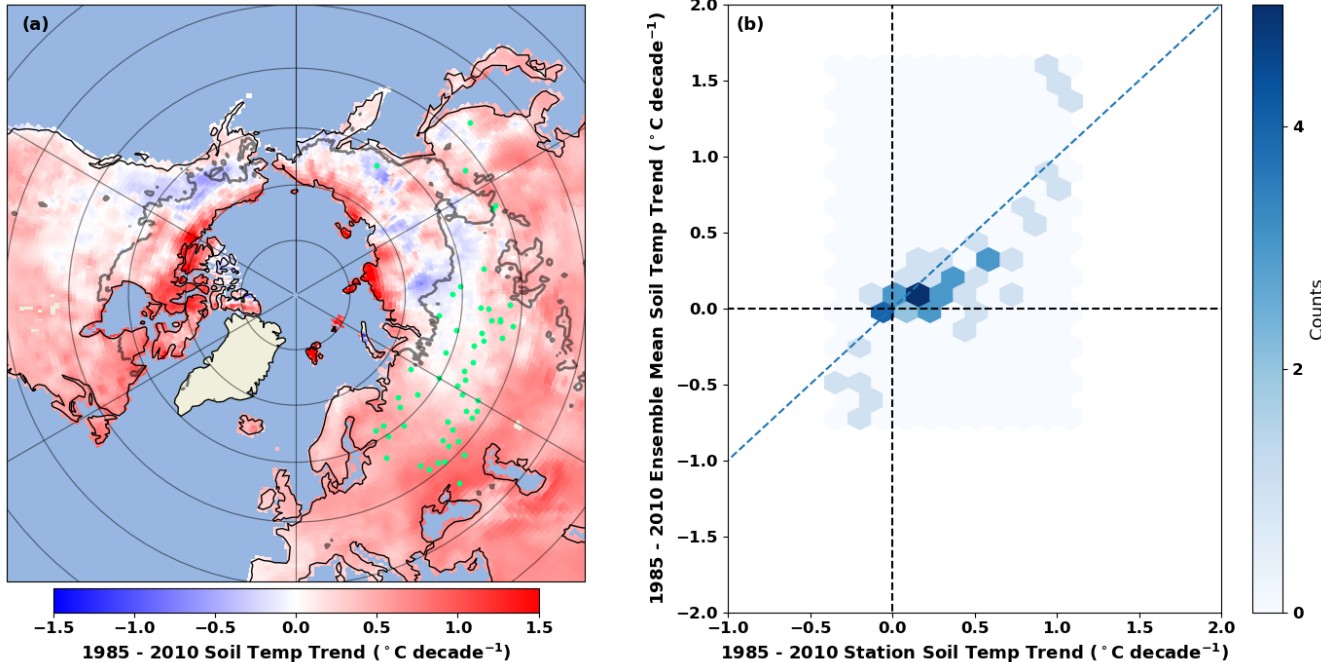

**Figure 12.** (a): 1985-2010 Ensemble Mean decadal soil temperature trends, near the surface, with the locations of validation grid cells included in the trend analysis shown as green dots. (b): Relationship between the near surface ensemble mean and station soil temperature trends (per decade). The black line represents the boundary of the permafrost zone (regions with at least 50% permafrost cover).

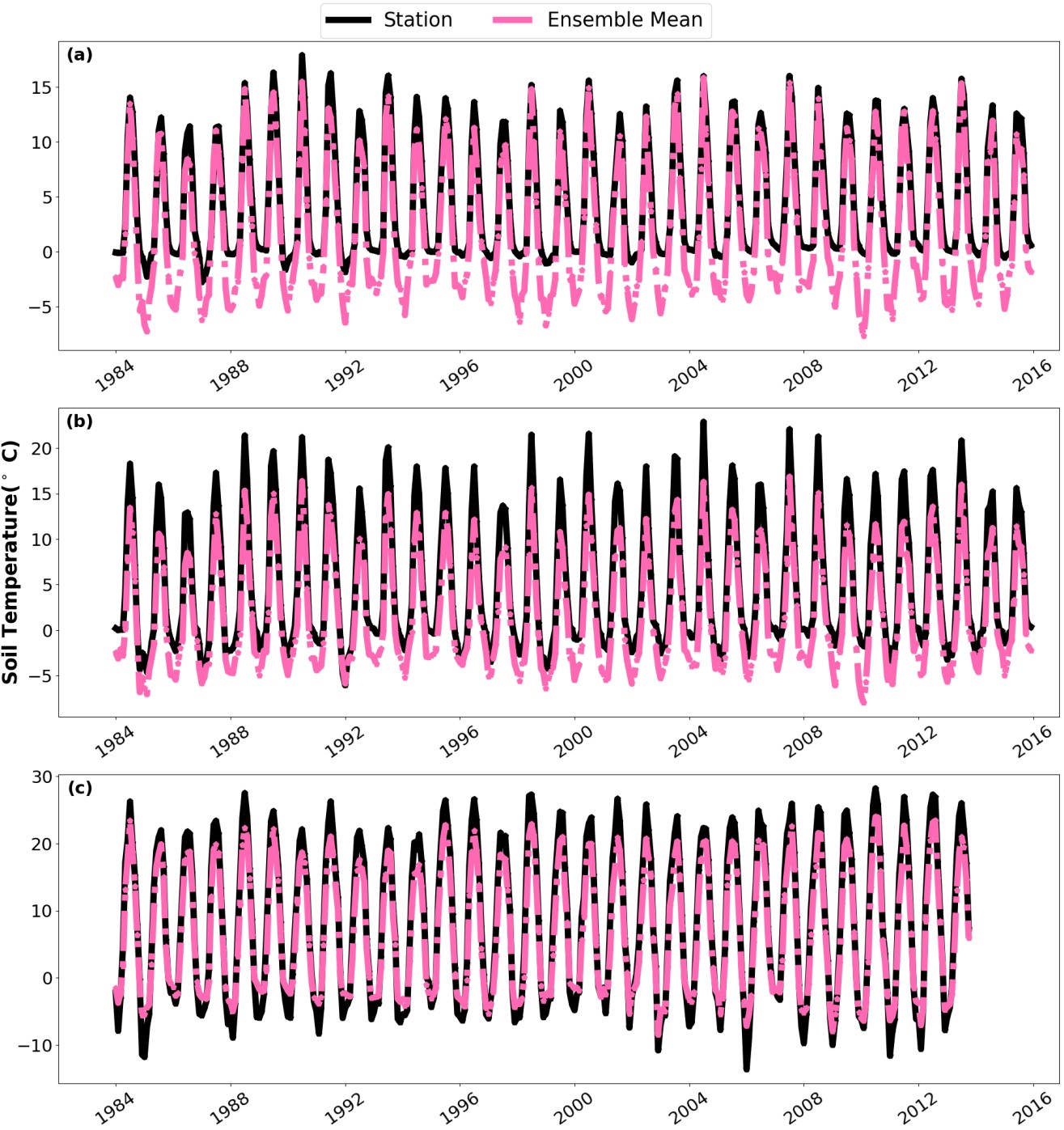

**Figure 13.** Timeseries from selected grid cells showing the ensemble mean (pink) and station (black) soil temperatures. (a): Timeseries where the ensemble mean simulates a winter minima that is too cold. (b): Timeseries where the ensemble mean simulates a summer maxima that is too cold. (c): Timeseries where the ensemble mean underestimates the seasonal cycle of soil temperatures.

## 6 Discussion and Conclusions

This study conducted a validation of pan-Arctic soil temperatures for eight reanalysis products, and validated a new ensemble mean pan-Arctic soil temperature dataset. The results are qualitatively similar to the findings of previous studies exploring reanalysis soil temperature performance in the extratropical northern hemisphere, which generally highlighted a cold bias in most products (Hu et al., 2019; Qin et al., 2020; Wu et al., 2018; Xu et al., 2019; Yang and Zhang, 2018; Zhan et al., 2020). Similar to (Li et al., 2021), we note greater biases in cold season soil temperatures, and our results qualitatively reflect the findings of Cao et al. (2020), who found that ERA5-Land exhibited warm soil temperature biases - particularly over higher latitudes.

The soil temperature trends reported here of a similar magnitude to those reported by Biskaborn et al. (2019), who found that permafrost soil temperatures generally warmed at a rate of $0.39° C \pm 0.15° C$ decade$^{-1}$, though ours differ in that they represent the mean extratropical northern hemisphere north of 40° N, whereas Biskaborn et al. (2019) predominantly focus on permafrost regions.

Other major differences here are that we develop an ensemble mean soil temperature product, and had a greater focus on higher latitude regions than most other studies. We also note a strong difference in seasonal performance. Relative to the warm season, the cold season is generally characterized by lower skill, larger near surface temperature biases, a larger spread in the reanalysis products' soil temperature variability and lower correlations with in situ soil temperatures. When all depths and seasons are considered, the ensemble mean product performs better than any individual product, exhibiting a consistently high skill, realistic soil temperature variability, and relatively small biases over all seasons.

Here we show an approximate estimate of the magnitude of soil temperature uncertainty associated with instrumental uncertainties, and those associated with structural differences and parameterizations in land models, using the standard deviation in soil temperature across time and as a function of station temperature. A complete quantitative assessment of the contributions of various sources of uncertainty is not possible using this dataset, as time-series did not have a consistent start or end date and consequently, the metrics are calculated using different climatologies across different grid cells. A more complete uncertainty analysis is beyond the scope here, but in the future could be achieved by limiting analysis to a subset of grid cells with consistent timeseries; for example by focusing on soil temperature networks such as the Michigan Enviro-weather Network, or the North Dakota Mesonet Network, or limiting the uncertainty analysis to a smaller portion of the permafrost region.

We find that the median spread in the spatially averaged soil temperature of stations in a grid cell is approximately 1.49° C (Figure 1, Panel B) – several degrees smaller than the standard deviation of reanalysis soil temperatures for a given station soil temperature; particularly over frozen soils (Figure 4). For example, when soil temperatures are below –20 ° C, soil temperature standard deviations increase to near 10° C in several products. Reanalysis two metre air temperatures maintain a relatively consistent standard deviation between 1.25° C to 1.75° C for most products, and only increase slightly to between 2.25° C and 3.5° C over the coldest station air temperatures (not shown). Unlike with soil temperature, the spread in reanalysis two metre air temperatures only increases modestly over the cold season (not shown). This would suggest that the largest degree of

uncertainty in reanalysis soil temperatures over the Arctic is most likely contributed by differences in the land models between
460 products, rather than from uncertainties in observed soil temperatures, or from differences in product air temperatures.

## 6.1 Uncertainties Associated with Land Model Parameterizations and Structural Differences

Uncertainties in soil temperatures associated with structural differences and parameterizations in land models can be grouped
into several categories. The first surrounds the simplified parameterizations controlling frozen soil processes. For example in
the Noah LSM - utilized by CFSR and FLDAS, freeze-thaw processes are highly simplified, and unsuited for permafrost sim-
465 ulations (Hu et al., 2019) - and may have contributed to the relatively large soil temperature biases simulated in these products.
Even the best performing products: ERA5 and ERA5-Land, are unsuited for simulation of permafrost soil temperatures, as
they fail to simulate phase-dependent changes in soil thermal conductivity (Cao et al., 2020).

Yang et al. (2020) noted that larger soil temperature biases over the Qinghai-Tibetan Plateau in deeper soil layers were likely
related to the fact that soil temperatures are less constrained by air temperature observations (and soil properties). This could
also explain why soil temperature biases in the warm season are larger at depth than near the surface in this study. Moreover, the
near surface soil layers tend to fall within the active layer (which undergoes seasonal freeze/thaw), while deeper soil layers are
more likely to contain permafrost. Permafrost has a high degree of impermeability, which prevents soil water from infiltrating
below the bottom of the active layer, and owing to latent heat considerations, leads to soil water freezing at the base of the
active later (Zhao et al., 2000), however these processes are not well represented in reanalysis LSMs Yang et al. (2020); Hu
et al. (2019).

LSMs such as the Simple Biosphere Model (used in JRA55), that use the force restore method to estimate soil temperature,
are prone to overestimating diurnal soil temperature range (Gao et al., 2004; Kahan et al., 2006), as well as the seasonal cycle
of soil temperatures (Luo et al., 2003). This is because they underestimate heat capacity, and overestimate temporal variation in
ground heat flux compared to more complex land models (Hong and Kim, 2010). Moreover, the force restore method assumes
a strong diurnal forcing from above, an assumption that is likely violated when snow cover is present (Tilley and Lynch, 1998;
Slater et al., 2001), because snow cover leads to a decoupling of the surface forcing from the soil below. These factors may
help explain why JRA55 is unable to simulate near surface soil temperatures as accurately as the other products explored in
this study explicitly incorporate representations of soil heat flux between soil layers (Niu et al., 2011; Koster et al., 2000;
van den Hurk et al., 2000; Balsamo et al., 2009), and hence they are able to simulate a dampening of seasonal variability of
485 soil temperatures at depth (and greater variability near the surface).

Burke et al. (2020) note that differences in snow cover properties were important in explaining soil temperature biases of
several Coupled Model Intercomparison Project 6 (CMIP6) models, and it is likely that differences in snow cover properties
between the LSMs studied here could account for some of the observed spread - particularly in the cases of ERA-Interim, ERA5
and ERA5-Land, because during the warm season, these products have similar soil temperature biases, but their performance
varies widely during the cold season (Figures 2 and 10), in large part because of snow density biases (Cao et al., 2020; Gao et al.,
2022). In ERA-Interim, the large cold (negative) bias during the cold season is strongly related to the fact that it overestimates
the observed snow density (Gao et al., 2022), and consequently also overestimates the thermal conductivity of the snow pack.

Conversely, snow density (and thermal conductivity) in ERA5-Land (and ERA5) are too low, and hence biases in snow density are a large contributing factor to the warm (positive) bias during the cold season (Cao et al., 2020).

Snow was also cited as a major controlling factor in soil temperature biases in ECMWF's Integrated Forecast System, which also uses the HTESSEL land surface model (Albergel et al., 2015). In the case of the Noah LSM, which is included in CFSR/CFSv2 and FLDAS, Li et al. (2021) found that an overestimation of snow cover was a major contributor to larger soil temperature biases in winter over the Qinghai-Tibetan Plateau. Shukla et al. (2019) and Shukla and Huang (2020) found that overestimation of snow cover in CFSR during autumn and early winter leads to an overactive snow-albedo feedback, and excessive cooling of the near surface soil layers. This translates into a strong cold bias at depth over the spring and summer, and likely explains why CFSR's warm season bias and RMSE at depth are the largest of all seven products examined in this study (Figure 2).

## 6.2  Uncertainties Associated with Scale Effects

Here we evaluated soil temperatures at a relatively coarse resolution of 0.75°. As such, it is difficult for reanalysis products to capture local scale variability in soil temperature. The sub-grid scale variability in soil temperatures calculated in Figure 1, Panel B is of a similar magnitude to those calculated by previous studies exploring sub-grid scale variability in cryospheric soil temperatures (Gubler et al., 2011; Morse et al., 2012; Gisnås et al., 2014), though are generally smaller than those reported by Cao et al. (2019). We found that the spatial variability in soil temperatures in one high latitude grid cell is larger than 10° C at times (Figure 1) - of a similar magnitude to those reported by Cao et al. (2019).

Moreover, as many grid cells in Eurasia only included a single in situ station, there is a large chance that this single in situ station may not necessarily be reflective of conditions elsewhere in the grid cell (Gubler et al., 2011). When multiple in situ stations were available, we took the spatial mean of all stations, in an attempt to estimate the mean soil temperature over the grid cell.

## 6.3  Uncertainties Arising from Sampling Variability

As was described in Section 5.2, the presence of missing data created a challenge for calculating in situ soil temperature averages. While most grid cells in Eurasia had relatively consistent timeseries, and fewer issues with missing data, this was not the case over North America. Rather than limit our analysis to a small number of grid cells with little to no missing data (as we did for the calculation of soil temperature trends), we chose to make use of all available data at each timestep when calculating our validation metrics (bias, RMSE, standard deviation, correlation and skill score). Thus, the spatially averaged in situ soil temperature did not always contain a constant number of depths or grid cells at each timestep in many grid cells over North America.

From Figure 1, Panel B, it is apparent that the median variability of soil temperatures between stations within a grid cell (spatial variation), 1.49° C, is roughly 1.75 times larger than the median variability of soil temperatures at different depths, 0.84° C, for a particular station (depth variation). Thus, it would appear that fluctuations in the number of stations comprising the spatially averaged soil temperature are responsible for a greater proportion of the uncertainty than fluctuations in the

number of depths included. However, it is also apparent that the uncertainties arising from variations in the number of grid cells included in a station average are substantially smaller than the spread between reanalysis products. During the cold season, the uncertainty in soil temperatures associated with the spread between reanalysis products is often two to three times larger than the uncertainty arising from fluctuations in station availability.

## 6.4 Applications and Future Work

The ensemble mean data product provides gridded, monthly-averaged soil temperature estimates of near surface, and deeper soil temperatures at a 0.75° resolution. Therefore, it is most suitable to regional or hemispheric-scale analyses of soil temperature climatologies, or their seasonal cycle, or to explore recent trends in soil temperatures. The product could also be used to provide boundary conditions for models that require soil temperature inputs, such as hydrological models, and for the validation of model soil temperatures. While the ensemble mean product still exhibits substantial cold biases over permafrost regions, and therefore is likely unsuitable for permafrost modeling, the RMSE over the permafrost zone of the ensemble mean product outperforms the RMSE of the best performing product by 0.5° C, on average (Figure S1) in the cold season, and hence it may still provide some added value for estimation of high latitude soil temperatures relative to the individual products.

A robust ensemble mean can be computed with four products (Figure **??**), which means a higher resolution ensemble mean data product could be created using a subset of higher resolution reanalysis products. For example, ERA5, ERA5-Land, MERRA2, and CFSR have resolutions lower than 0.5 degrees. Using a similar blending methodology, we have been investigating the performance of a 0.31-degree product (using a smaller subset of products that provide data at higher spatial resolution). We have also performed similar analyses with a 0.05-degree soil temperature product, using interpolated soil temperatures from the Arctic System Reanalysis version 2 (ASR), ERA5-Land, and FLDAS. The goal has been to assess the impact of spatial resolution on performance of the ensemble mean product. We are hoping to include these results in a follow-up paper. Future work should aim to investigate how differences in snow cover and snow density between the reanalysis products may influence biases in the individual products. On a related note, future studies should also emphasize how differences in the land model structure and parameterization may account for the spread in soil temperatures.

*Data availability.* The CRU TS v 4.07 2m air temperature dataset can be found on the CRU TS dataset website. GTN-P data (GTN-P, 2018) is available from The Global Terrestrial Network for Permafrost, while the Kropp et al. (2020) dataset is available from Heather Kropp's Arctic Data Center page. Russian Hydromet (Sherstiukov, 2012) data are available from RIHMI-WDC, while Nordicana data can be obtained from Nordicana D. CFSR (Saha et al., 2010), CFSv2 (Saha and Coauthors, 2012), ERA-Interim (European Centre for Medium-Range Weather Forecasts, 2012), and JRA-55 (Japan Meteorological Agency, 2014) data were obtained from the National Center for Atmospheric Research (NCAR)'s Research Data Archive (RDA). FLDAS (McNally and NASA/GSFC/HSL, 2018), and MERRA2 (Global Modeling and Assimilation Office, 2015) were obtained from the Goddard Earth Sciences Data and Information Services Center (GES DISC). ERA5 (European Centre for Medium-Range Weather Forecasts, 2019), and ERA5-Land data (Muñoz-Sabater, 2019) was downloaded from the Copernicus Climate Change Service (C3S) Climate Data Store (CDS). The ensemble mean soil temperature dataset has been made available on the Arctic Data Center.

*Author contributions.* TH and CGF conceived of the study, TH gathered, and analyzed the data, and TH, CGF interpreted the data. HK provided in situ data to the study, and TH, and CGF wrote the manuscript, with contributions from HK.

*Competing interests.* The authors declare that they have no known conflicts - financial or personal, that could have appeared to influence this work.

*Acknowledgements.* This work was funded under a National Science and Engineering Research Council (NSERC) PGS-D scholarship. The authors would like to thank the anonymous referees, as well as Dr. Hugh Brendan O'Neill and Dr. Andre Erler for their helpful comments.

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
