# Peer review of "Validation of Pan-Arctic Soil Temperatures in Modern Reanalysis and Data Assimilation Systems"

_The Cryosphere, 2022_

## Author Comment (AC1)

[Figure]

*Figure R1. Permafrost zonation at validation grid cell locations based on (left) Smith and Riseborough (2002) mean annual air temperature method and (right) Obu et al. (2019).*

[Figure]

*Figure R2. Scatterplot of product soil temperatures relative to station soil temperatures. The line bisecting the panels referes to the 1:1 line. Points above the line are observations where the reanalysis product is too warm, while points below the line refer to cases where the reanalysis product is too cold.*

---

## Author Comment (AC2)

[Figure]

*Figure R1. Validation grid cells for the near surface (left) from pre-print and (right) updated data set in revised article. The preprint included 24 validation grid cells in North America, and 244 over Eurasia. The updated dataset now includes 135 validation grid cells over North America (and 247 over Eurasia). Moreover the number of grid cells over the permafrost zone in North America (30 grid cells) and Eurasia (45 grid cells) are also reasonably similar. Note that here we are using the Obu et al. (2019) permafrost distribution (and not MAAT) to classify permafrost.*

[Figure]

*Figure R2. Relationship between soil temperature standard deviation and mean annual air temperature (MAAT) for DJF over the 1991-2010 climatological period.*

---

## Author Response (AR1)

**Author Point-By-Point Response**

**Reviewer #1**

The authors would like to thank Referee 1 for their helpful comments. We have substantially restructured the Results (Sections 4 and 5) and the Discussion/Conclusion (Section 6) to shorten its length and tighten the discussion. Section 5 has been modified to an investigation of variability and trends in the ensemble mean soil temperature product compared to a subset of stations with longer temperature records.

**Major Comments**

1) Reformulate and shorten the manuscript (maybe as a brief communication) with a very specific focus on soil temperature validation

We have substantially shortened the results sections (Sections 4 and 5) and have changed the section on the ensemble mean soil temperature climatology (Section 5.2) to focus on the validation of variability in soil temperature extremes (winter minimum and summer maximum soil temperatures), and soil temperature trends over a subset of grid cells with longer temperature records.

In response to the structural changes in the text, we've also made changes to a number of the figures, and their ordering has changed in some cases:

- Figure 1, Panel A has been updated substantially to account for the new grid cells added, and changes in Section 5.2 (mainly the delineation of the subset of grid cells used for soil temperature trend analysis). Panel B has been updated to show the relative importance of spatial variability and depth variation in soil temperature.
- Figure 2 now includes a new metric – a Taylor (2000) type skill score metric providing an objective estimate of the overall performance of each product
- Former Figure 3 has been renumbered as Figure 6
- Figure 4 has been revamped to only include the relationships between the station soil temperature and the product soil temperatures
- Former Figure 5 is now Figure 3
- Former Figure 6 is now Figure 5
- Former Figure 7 is now Figure 8
- Figure 7 is now a spatial map of the Ensemble Mean soil temperature biases)

- In section 5.2, former Figure 8 has been removed, and instead 3 new figures have been added (Figures 9 – 11).

2) In Sec. 4 & 5, the authors present the evaluation results together with a large part of the discussion, and additional discussions are given in Sec. 6. This makes the manuscript very unclear and difficult to follow.

The authors have separated out discussion material from the Results (Section 4 and 5). Relevant discussion material has now been incorporated into a substantially revised Section 6 that now includes a brief summary of how the results of this study relate the findings of previous work.

3) The discussion in Sec 6 is very general and superficial, and is not tightly connected to previous sections. For instance, the gap of site-scale observation and model grid (about 10–100 km), or so-called scale effects, is widely reported. P23, L392–398, this part is very confusing. Does the misclassification of permafrost affect the results? Please make sure only to present the most relevant parts here to avoid diluting your real contributions.

Section 6 discussion was reformulated to incorporate relevant discussion material originally included in Sections 4 and 5. The section begins with a summary of the key results and how these relate to previous studies' findings. The discussion now focuses around four main themes:

- Uncertainties associated with land model parameterizations and structural differences in the land models
- Impacts of discontinuities in the reanalysis timeseries
- Uncertainties associated with scale effects
- Uncertainties arising from sampling variability

The discussion is more focused as the authors clearly emphasize how these uncertainties could potentially influence the results.

4) The authors presented and discussed the reanalysis soil temperature deviation. I am wondering why this is important here and how this could be used for validation purposes? The strong variation of soil temperature in the cold season could be expected due to the presence of a snow layer, see Figure 6 from Burke et al., (2020).

The authors thank the reviewer for bringing this figure to our attention. We use the normalized standard deviation as a measure of the temporal variance of soil temperatures across the grid cells. Doing allows us to assess the range of simulated soil temperatures at each grid cell for a particular product, to see if it can capture a similar seasonal cycle of temperatures to that of the observations.

When we describe soil temperature variability in the cold season we are referring to two main features – first that the individual products themselves show a larger variance in soil

temperatures than they do during the warm season. Second, we are also describing the spread in soil temperature variance between products. It is likely that differences in snow cover properties may help account for the latter, and we have added a reference to Burke et al. (2020) in Section 6.1 beginning at line 432:

> Burke et al. (2020) note that differences in snow cover properties were important in explaining soil temperature biases of several Coupled Model Intercomparison Project 6 (CMIP6) models, and it is likely that differences in snow cover properties between the land models of the reanalysis products could account for some of the observed spread - particularly in the cases of ERA-Interim, ERA5 and ERA5-Land.

5) The climatology based on the ensemble results is somehow unfocused. The purpose of this study is "validation of pan-Arctic (and Boreal) soil temperatures from eight reanalyses and land data assimilation system (LDAS) products." (see P2, L53–54), rather than analyzing the climatology. To be more focused, authors could compare and evaluate the trend of ensemble results with site-scale observations.

We agree that a focused evaluation of the trends against a subset of the stations with longer timeseries is of value and have restructured Section 5.2 to focus instead on validating decadal soil temperature trends against a subset of station estimates with longer timeseries Figure 11 and Supplemental Figure 6). We also validate the annual soil temperature minimum and maximum soil temperatures (Figure 10 and Supplemental Figure 5) in this section.

6) P8, L180: Then why not directly use the IPA map? You could also find the global permafrost zonation index map from Gruber et al., (2012).

The authors thank the reviewer for this suggestion. We have incorporated the Obu et al. (2019) permafrost zonation index map into our analysis, and the permafrost zone now refers to regions with at least 50% permafrost cover, while the zone with little to no permafrost refers to regions with less than 50% permafrost cover. The contour lines on Figure 1, Panel A encircle regions with at least 50% permafrost cover – as estimated by Obu et al. (2019). The overall conclusions of the study were not impacted by this change.

**Minor Comments**

1) P2, L24: Permafrost carbon and climate warming loop are complex, and thus ...could act as a "possibly/potentially" positive…

We have revised the sentence to read as: "Continued warming, and thawing of permafrost soils, and related decomposition of carbon could act as a **potential** positive feedback on warming, by releasing more methane ($CH_4$) and carbon dioxide ($CO_2$) into the atmosphere."

> 2) P2, L31: Qinghai-Tibetan **P**lateau

Corrected.

> 3) P2, L45–49: Ensemble simulation has also been used for permafrost simulation, for instance, Cao et al., (2019), although these studies do not directly use the soil temperature

We have added a reference to Cao et al. (2019) in our introduction - in the following sentence:

"Ensemble mean datasets based on combinations of in situ, model, satellite and reanalysis data have been used to reduce biases in estimates of snow water equivalent (Mudryk et al.,2015), soil moisture (Dorigo et al., 2017; Gruber et al., 2019), precipitation (Beck et al., 2017, 2019), **as well as for local scale permafrost simulations (Cao et al., 2019)**."

> 4) P4, L122: The variation of soil temperature is complex and typically depends on surface condition (i.e., snow layer, vegetation), soil properties (i.e., soil organic content), and soil depth. It could vary very large at the hourly and daily scales.

The authors thank the reviewer for this helpful comment. As we are using soil temperatures averaged between 0cm and 30cm in the near surface, and between 30cm and 300cm at depth, we had presumed that the soil temperatures should show reduced variation on daily and hourly timescales. We have revised this sentence to read "Many of the in situ (station) sites reported measurements at hourly or daily frequency, however we chose to perform the analysis at monthly time scales, in order to focus on processes controlling the seasonal cycle of soil temperatures."

> 5) P6, L135: How much the difference could be? Could you please write it down?

The authors presume that the reviewer is asking by how much the soil temperatures may vary between stations within a grid cell. Panel B of Figure 1 gives an estimate of the variability of soil temperatures within a grid cell – The median spatial standard deviation is ~2°C, however soil temperatures may vary by as much as 13°C in the case of a couple of high latitude grid cells. The variability is of a similar magnitude to previous studies exploring sub-grid scale variability in cryospheric soil temperatures (e.g. Gubler et al., 2011; Morse et al., 2012; Gisnås 2014; Cao et al., 2019).

> 6) P6, L141: ...2 to 12..

We chose to write 2 as "two", since style conventions in The Cryosphere specify that all single digit numbers (unless they are followed by units) should be written as a word.

7) P6, L142: The so-called "scale effects" has been widely reported, see Gubler el al., (2011) for the Alps and Cao et al., (2019) for high latitudes. Please cite relevant references.

We thank the reviewer for pointing out several relevant references on scale effects. In our revisions, we have linked our estimates of scale effects with those in the literature, and show that our results qualitatively agree with those exploring scale effects in seasonally frozen and permafrost soils (e.g. Gubler et al., 2011; Morse et al., 2012; Gisnås 2014; Cao et al., 2019).

We have added the following sentences in Section 6.3 to highlight the impacts of scale effects:

> The sub-grid scale variability in soil temperatures calculated in Figure 1, Panel B is of a similar magnitude to those calculated by previous studies exploring sub-grid scale variability in cryospheric soil temperatures (Gubler et al., 2011; Morse et al., 2012; Gisnås et al., 2014), though is smaller than those reported by Cao et al. (2019). If the strict requirements surrounding consistency in the number of stations and depths are relaxed, allowing for stations in permafrost regions to be included, spatial variability in soil temperatures is larger than $10 \circ$ C at times in a couple of high latitude grid cells (not shown) - similar to the findings of Cao et al. (2019).

Corrected.

Revised.

Corrected.

11) P16, L256: Zero curtain period is heavily dependent on the soil moisture rather than the active layer thickness

The authors thank the reviewer for this helpful comment. We have made substantial revisions to the Results, and this paragraph is no longer present in the manuscript. We've moved our discussion of freeze-thaw parameterizations to Section 6, for example.

Fixed.

13) Table 1: Could you please also add the soil discretization information here, such as depth for each layer and the total soil column depth? Please double-check the spatial resolution of all the reanalyses, ERA5 should be 0.25°, ERA-Interim is 0.75°, and MERRA-

2 is 0.5°×0.625°. Depending on the datasets you used, JRA-55 is 1.25° for the reanalysis level and 0.56° for the model level

We have corrected the spatial resolutions in Table 1. The information in Table 1 has also been split into 2 tables in order to include information about the depths of soil layers in each product.

14) Figure 4: Do you really need so many sub-plots? The inter-comparisons among different reanalyses are shown here but not discussed in the main text. Did I miss something important?  Please also add the 1:1 line, so that readers could clearly see the cold/warm bias

The most important comparisons to be made here are the performances of the individual products against the station – the outer margins of Figure 4 in the paper – which is the focus of the text. We also used the histograms of the warm/cold season to look at the variability of soil temperatures in the warm and cold season. We have revised Figure 4 such that it now only shows the relationship between station soil temperatures and each of the products, for both depths.

15) Figure S3: Could you please improve the resolution of Figure S3?

We have recreated Figure S3 (Figure S5 in the revised manuscript) and it should be at a higher resolution now.

**Reviewer #2**

The authors would like to thank Referee 2 for their helpful comments. As a part of our revisions, we have gathered substantially more data for North America, and have recalculated all metrics. In our updated database, we now have 135 validation grid cells over North America; 30 of which are located over the permafrost region. By utilizing soil temperature data from a variety of hydrometeorological and agricultural monitoring networks, our dataset now provides the most comprehensive analysis to date of soil temperatures across northern and southern Canada and the Great Lakes basin.

**Major Comments**

1. The authors list potential future applications of the ensemble mean product, but I would wish to see a bit more discussion on its current usability, given that the recorded biases remain quite high and display some regional patterns. The underlying reasons for these are addressed in the manuscript but not how the biases would affect, e.g., permafrost simulations where a bias or RMSE of above 2° C can have notable implications.

We thank the reviewer for this comment. Several of the products have an RMSE of ≤4°C – particularly over permafrost regions (as shown in Figure S1). In most products, this is expressed as a cold bias, which would suggest that reanalysis products may overestimate permafrost extent and underestimate active layer thickness. The ensemble mean biases and RMSE are generally better than (or similar to) the best performing product, especially when all seasons and depths are considered. In addition, the ensemble mean soil temperatures show a more realistic pattern of soil temperature variability in the permafrost zone compared to the individual products themselves.

The ensemble mean product provides gridded, monthly-averaged soil temperature estimates of near surface, and deeper soil temperatures at a 1° resolution. Therefore, it is most suitable to regional or hemispheric-scale analyses of soil temperature climatologies, or their seasonal cycle, or to explore recent trends in soil temperatures (since 1980). The product could also be used to provide boundary conditions for hydrological models. In fact, a higher resolution version of this product (see our response to Question 12 in Minor Comments) is being used for such a purpose and will be described in a follow-up study.

The authors acknowledge that the ensemble mean soil temperature product would most likely yield an overestimation of permafrost extent, given that it is biased cold by 3-5°C, on average, at high latitudes. That being said, over permafrost regions, the RMSE of the ensemble mean product outperforms the RMSE the best performing product by ~2°C, on average, and hence it may still provide some added value for estimation of high latitude soil temperatures relative to the individual products.

We have added the following paragraph to the manuscript in Section 6.5 (Applications for the Ensemble Mean Product and Suggestions for Future Work):

> The ensemble mean data product provides gridded, monthly-averaged soil temperature estimates of near surface, and deeper soil temperatures at a 1° resolution. Therefore, it is most suitable to regional or hemispheric-scale analyses of soil temperature climatologies, or their seasonal cycle, or to explore recent trends in soil temperatures. The product could also be used to provide boundary conditions for models that require soil temperature inputs, such as hydrological models, and for the validation of model soil temperatures. While the ensemble mean product still exhibits substantial cold biases over permafrost regions, and therefore is likely unsuitable for permafrost modeling, the RMSE of the ensemble mean product outperforms the RMSE of the best performing product by ~2°C, on average, and hence it may still provide some added value for estimation of high latitude soil temperatures relative to the individual products.

2. At places the text is hard to follow (especially Section 4.3, see detailed comments below) owing to the multiple simultaneous comparisons: near surface vs. at depth soil temperatures, cold season vs. warm season, permafrost vs. no to little permafrost, North America vs. Eurasia, and DJF vs. JJA. I suggest the authors to make sure all sections are clearly defined.

Section 4 has been substantially restructured. We begin in Section 4.1, where we discuss the extratropical northern hemisphere mean results, before moving to discuss differences between the warm and cold seasons in Section 4.2. Regional differences are discussed in Section 4.3, and we have taken care to separate our discussions of permafrost presence from our comparisons between the continents, especially now that we have a greater number of validation grid cells over North America (making the comparisons between Eurasia and North America more meaningful).

3. L104: The authors suggest that their study is *"To the authors' knowledge, this one of the first studies to compile a comprehensive set of in situ soil temperature measurements across the Eurasian and North American Arctic, from multiple diverse sparse networks".* While it may be true that this is true for the "one of the first" part, it should be noted that the compilation is not totally novel, given that similar in situ temperature datasets have been compiled not only by Cao et al. (2020, in the references) but also, e.g., by Karjalainen et al. (2019) and Ran et al. (2022) who used mostly the same data sources, albeit computing temperatures averages for a much larger depth (several meters deep in permafrost but also in non-permafrost soils). Moreover, Lembrechts et al. (2020) have published a global soil temperature compilation of soil and near-surface temperatures. I suggest the authors to consider if their statement needs some elaboration, e.g., does the compiled dataset differ from previous datasets in some ways.

The authors recognize the notably different sampling size for North America but retain from explaining why no more data were collected, apart from mentioning the overall data scarcity in northern Canada, to correct the imbalance between North America and Eurasia. Based on the previous data compilations (see above), there should be suitable measurement time series available from North America.

The authors thank the reviewer for making us aware of these studies. As a result, the biggest change in the revised manuscript is the inclusion of a large amount of new soil temperature data from North America. Figure R1 compares the previous and updated distribution of validation grid cells, which now contains 135 validation grid cells over North America near the surface; 30 of which are located over the permafrost region. This means that our sample of sites for North America is now more comparable to the 247 grid cells in Eurasia (45 of which span the permafrost region). The much improved coverage across North America is evident in Figure 1, Panel A:

[Figure]

The new data are drawn from multiple sources, and we reiterate our claim from the original manuscript that this collection of pan-Northern Hemisphere soil temperature data constitutes a novel and important contribution to the permafrost research community. Over the permafrost region, we've assembled data from the Yukon (Yukon Geological Survey, 2021) and the NWT (Cameron et al., 2019; Ensom et al., 2019; Gruber et al., 2019; GTN-P, 2018; Spence and Hedstrom, 2018a; Spence and Hedstrom, 2018b; Street, 2018).

In addition, we have incorporated data from several soil monitoring and hydrometeorological networks across Southern Canada and the Great Lakes basin of the United States, that, to our knowledge, are not included in any of the above papers. These include 85 stations from the Manitoba Mesonet network (RoTimi Ojo and Manaigre, 2021), 83 stations in Michigan and western Wisconsin (MAWN, 2022), 31 stations from the Alberta Climate Information Service network (Alberta Agriculture, Forestry and Rural Economic Development, 2022), and 150 stations from North Dakota (NDAWN, 2022). We are also including data from a peatland ecosystem in Metro Vancouver (Lee et al., 2017; Lee et al., 2021), as well as data from 11 stations in central and Northern BC (Déry, 2017; Hernández-Henríquez et al., 2018; Morris et al. 2021), and 2 stations in southern Quebec (Arsenault, 2018; Fortier, 2020).

We have also been in contact with the data providers from the Real Time In-Situ Monitoring Network (RISMA), however the data was not available to include at the time this response was submitted. We hope to include the RISMA dataset (which includes 13 stations in southern Manitoba, 6 stations in southeastern Ontario, and 4 stations in southern Saskatchewan) in follow-up studies.

While the Ran et al. (2022) study included borehole measurements from southern Canada, the data did not include information about the seasonal cycle of soil temperatures. Thus, our work presents the most comprehensive analysis to date of soil temperatures across northern and southern Canada and the Great Lakes basin.

**Minor Comments**

1. LL140-141: *"Panel B of Figure 1 shows the spatial standard deviation of monthly surface soil temperatures for grid cells with more than two stations included."* However, in Figure 1b, grid cells with two stations are also shown. Also, I remain unsure whether there are any grid cells with >1 stations in Eurasia?

*Based on the grid cells that met our criteria for validation, there were no grid cells in Eurasia with two or more stations included. A clarification has been added to the text:*

> *Over Eurasia, grid cells contained a single in situ measurement location.*

*Figure 1 has been revised as part of the changes that were made to Section 5.2 (Trends and Variability in Seasonal Extremes in the Ensemble Mean Product). Panel B now compares the average temperature variation between stations within a grid cell (spatial variability) with the average variation in soil temperatures across the top 30cm at a*

*particular station, for a subset of grid cells that have a consistent number of stations and depths in their timeseries.*

[Figure]

2. L236: Reference should be to Fig. S1, right?

L236 mentions that "several factors may explain the increased variability in soil temperatures over permafrost regions." We presume that you may have meant L226, which describes the difference in the mean bias/RMSE over North America versus Eurasia?

In the original manuscript, Figure S1 displays the mean bias and RMSE over the combined Pan-Arctic permafrost zone. Here we meant to refer to Figure S2, which showed the difference in bias between Eurasia and North America.

We have since made a change so that Figure S1 now refers to the bias and skill score of products in permafrost regions versus those outside the permafrost zone:

[Figure]

*Figure S1. Bias (bar plot) skill scores (scatter) for the cold season (≤-2°C) and the warm season (> -2°C) over the permafrost zone (Panels A and C) and the zone with little-to-no permafrost (Panels B and D) . The top panels display the bias and RMSE for the near surface (0cm - 30cm) layer, while the bottom panel displays the bias and RMSE at depth (30cm - 300cm). Models are ordered based on cold season RMSE (from the smallest to largest). Products are listed in alphabetical order, with the ensemble mean listed at the end for comparison.*

**Figure S2 now shows the same metrics, but for Eurasia and North America:**

[Figure]

*Figure S2. Bias (bar plot) skill scores (scatter) for the cold season (≤-2°C) and the warm season (> -2°C) over North America (Panels A and C) and Eurasia (Panels B and D) . The top panels display the bias and RMSE for the near surface (0cm - 30cm) layer, while the bottom panel displays the bias and RMSE at depth (30cm - 300cm). Products are listed in alphabetical order, with the ensemble mean listed at the end for comparison.*

3. L239: What correlations, the ones between measurements and reanalysis temperatures? A slight elaboration would help the reader to see that what are compared in the sentence.

Yes, we were referring to correlations between the observed soil temperatures and the reanalysis temperatures. Section 4 has been substantially revised, and we have taken care to clarify that correlations are referring to those between reanalysis soil temperatures and station soil temperatures.

4. LL240-241: I also struggled with this sentence. What is the opposite situation here? It is hard to follow the comparisons between permafrost and little to no permafrost, as well as near-surface and at depth temperatures at the same time, especially since the results are not shown.

Here we were referring to the fact that the reanalysis products are more likely to overestimate the observed variance over the permafrost region at depth. The results section has been substantially altered in a way that we believe is easier to follow (see our response to comment #3 for further details).

5. LL243-246: Are these results related to the permafrost binning? It's fine if they are not, but overall Section 4.3 is at times hard to follow because it deals with both permafrost binning and regional comparisons.

We have revised Section 4.3 such that differences between permafrost/no permafrost and regional differences are more clearly separated. We begin with a comparison of the differences between regions with permafrost, and regions outside the permafrost zone, and then include a short paragraph at the end of Section 4.3 that explains differences over North America and Eurasia.

6. L405: Instead of the cold season standard deviations, should you not refer here to cold stations/observations? That is, figure 6 does not distinguish between warm and cold season.

What we were describing here is that when soil temperatures are frozen (and particularly for soil temperatures below –20°C), soil temperature standard deviations increase to near 10°C in several products. The sentence has been altered as follows:

> For individual products, the variability in reanalysis soil temperature for a given observed soil temperature (as measured by their standard deviation) is generally greatest over frozen soil conditions (particularly temperatures below -20°C) - further evidence of the reduced agreement between product soil temperatures and observations.

7. L261: The ensemble mean product is not properly addressed until deep into the results (validation) section. I suggest presenting the ensemble mean product and its calculation procedure already in the early stages (possibly inside section 2.1.).

As suggested, we have added a subsection in the Methods – Section 3.3 that explains how the ensemble mean soil temperature product was created, and the soil layers included from each product for the two depths.

8. L303: I find "coastal regions" not the ideal term here because the regions with the highest variability are far more than that. In winter, greatest variation associates with the coldest regions, yet not exclusively either. Could the variation here be related to snow cover duration / snow thickness as mentioned elsewhere in the text?

We agree that "coastal regions" does not adequately describe the spatial pattern of variability – particularly in winter – a more appropriate description would likely be "coldest regions". Figure R1 shows a scatterplot of the relationship between mean annual air temperature (MAAT) and soil temperature standard deviation, when soil temperature variance is largest. The figure shows that soil temperature standard deviation and MAAT have a moderately strong negative correlation of -0.69. Moreover, it appears that regions with extreme continentality (such as eastern Siberia) show the largest standard deviations. While it is possible that snow cover characteristics may be important in certain regions, a detailed snow cover analysis is beyond the scope of this paper – and will be the focus of a follow-up paper.

[Figure]

*Figure R1. Relationship between soil temperature standard deviation and mean annual air temperature (MAAT) for DJF over the 1991-2010 climatological period.*

Section 5.2 has been substantially altered and now focuses on validating soil temperature trends. When describing the ensemble mean temperature trends, we now say the following:

> The ensemble mean soil temperature dataset shows that most regions see small positive annual mean soil temperature trends of ≤ 1°C decade$^{-1}$, with slightly larger trends in the Canadian Arctic Archipelago and in Siberia, for example.

**Technical Corrections**

1. L61: Please, open the abbreviation GLDAS-CLSM already here.

We have expanded GLDAS-CLSM to read "Global Land Data Assimilation System – Catchment Land Surface Model" here:

> Products were remapped onto the Global Land Data Assimilation System – Catchment Land Surface Model (GLDAS-CLSM) grid for comparison, using three different methods: nearest neighbour, bilinear interpolation, and first-order conservative remapping.

2. LL80-83: Check grammar of the sentence. Maybe delete the word "that" at line 81?

This sentence should say "In ERA5, a weak coupling exists between the land surface and atmosphere. It includes an advanced LDAS that incorporates information regarding the near-surface air temperature, relative humidity, as well as snow cover (de Rosnay et al., 2014), along with satellite estimates of soil moisture and soil temperature from the top 1m of soil (de Rosnay et al., 2013)."

3. L191: Figure 2 does not have panels C and D.

This sentence should read "Warm season biases tend to be slightly larger at depth (Figure 2, Panel B) for most products (by 1°C – 2°C)."

4. Figure 3: This is a nice figure with lots of information in it. The letters in "Correlation coefficient" are clumped together and could be corrected.

We agree – Figure 3 (now Figure 6 in the revised manuscript) has been updated to correct the issues with "Correlation Coefficient" in the text.

5. Figure 4: Stratification of the values in histograms is not explained. Please add it to the caption.

This figure has been altered in the revised manuscript in response to other revisions, and the histogram no longer appears in the updated version.

6. Figure 5: Y-axis is a bit messy. Consider adjusting the interval at which temperatures are denoted.

We have altered Figure 5 (now Figure 3 in the revised manuscript) so that major ticks at every 2°C.

7. Figure 8: DJF missing from Panel A label.

Figure 8 has changed in response to other revisions and has been replaced.

8. L286: NH → northern hemisphere

Instances of NH have been changed to "northern hemisphere".

9. L290: Why are ensemble mean at depth temperatures not shown? Could be part of the supplement. Figure 9 also shows at depth results, so it would be interesting to

see how the models reconstruct frozen ground in JJA, although it is acknowledged that this is not explicitly representative of permafrost.

Our decision to not include the results at depth was because the pattern correlations were quite similar to those near the surface (with a pattern correlation of ~0.95 over the study area). The overall features were generally quite similar, however showing a smaller annual range of temperatures.

Section 5.2 has been substantially altered to focus on validation of annual mean soil temperature trends and performance in the winter minimum and summer maximum soil temperatures (Figures 9 - 11). While the section focuses on near surface performance, we have included a brief comparison with the performance at depth and have included equivalent figures as supplementary figures (Figure S5 and A6).

10. L366: Please put Gruber et al. 2018 inside parentheses.

This sentence has been removed as the Discussion section has undergone substantial revisions. We have ensured that citations are correctly formatted.

11. L369-370: *"Moreover, the impact of snow cover on soil temperature is generally more pronounced over permafrost regions (regions of seasonal frost)."* Is something missing here? Should it be "compared to regions of seasonal frost" or what is the idea?

We agree that this sentence is confusing. It should have read "Moreover, the impact of snow cover on soil temperature is generally more pronounced over permafrost regions relative to regions of seasonal frost."

Section 6 (Discussion and Conclusion) has undergone substantiative revisions in response to comments from Referee 1, so the sentence no longer appears in the updated manuscript.

12. LL418-419: Could you elaborate, what does it mean *"is being explored"*?

Using a similar blending methodology, we have been exploring the impacts of spatial resolution on soil temperature performance, using a subset of the products examined in this study. A 0.31-degree product based on CFSR, ERA5, ERA5-Land and GLDAS-Noah was explored, along with a 0.05-degree soil temperature product, using interpolated soil temperatures from the Arctic System Reanalysis version 2 (ASR), ERA5-Land, and the Famine Early Warning Systems Network (FLDAS).

We have included the following information in our revised manuscript:

Using a similar blending methodology, we have been investigating the performance of a 0.31-degree product (using a smaller subset of products that provide data at higher spatial resolution). We have also performed similar analyses with a 0.05-degree soil temperature product, using interpolated soil temperatures from the Arctic

System Reanalysis version 2 (ASR), ERA5-Land, and the Famine Early Warning Systems Network (FLDAS). The goal has been to assess the impact of spatial resolution on performance of the ensemble mean product. We are hoping to include these results in a follow-up paper.

13. L428: Please provide a url for the ensemble mean dataset on the ADC.

The original version we submitted had all URLs as hyperlinks. We see that the hyperlinks are not present in the version available online, so we have added a hyperlink to the ensemble mean dataset:

> The ensemble mean soil temperature dataset has been made available on the Arctic Data Center (https://doi.org/10.18739/A2RN3085P).

14. L583: Database title and url missing.

We have added a database title and URL for Heather Kropp's dataset.

> while the Kropp et al. (2020) dataset is available from https://doi.org/10.18739/A2736M31X.

**References:**

Agriculture and Agrifoods Canada. Real Time In-Situ Monitoring for Agriculture. (Retrieved: July 2022 from: https://agriculture.canada.ca/SoilMonitoringStations/historical-data-en.html)

Alberta Agriculture, Forestry and Rural Economic Development, Alberta Climate Information Service (ACIS). Current and Historical Alberta Weather Station Data. (Retrieved: July 2022 from: https://acis.alberta.ca).

Cao, B., Quan, X., Brown, N., Stewart-Jones, E., and Gruber, S.: GlobSim (v1.0): deriving meteorological time series for point locations from multiple global reanalyses, Geosci. Model Dev., 12, 4661–4679, https://doi.org/10.5194/gmd-12-4661-2019, 2019.

Cameron, E. A., Lantz, T. C., O'Neill, H. B., Gill, H. K., Kokelj, S. V., and Burn, C. R.: Permafrost Ground Temperature Report: Ground temperature variability among terrain types in the Peel Plateau region of the Northwest Territories (2011-2015), Northwest Territories Geological Survey, Northwest Territories, Canada, 2019.

Déry, S.: Cariboo Alpine Mesonet (CAMnet) Database,. Zenodo. https://doi.org/10.5281/zenodo.1195043 , 2017.

Ensom, T., Kokelj, S. V., and McHugh, K. K.: Permafrost Ground Temperature Report: Inuvik to Tuktoyaktuk Highway stream crossing and alignment sites, Northwest Territories, Northwest Territories Geological Survey, Northwest Territories, Canada, 2020.

Gisnås, K., Westermann, S., Schuler, T. V., Litherland, T., Isaksen, K., Boike, J., and Etzelmüller, B.: A statistical approach to represent small-scale variability of permafrost

temperatures due to snow cover, The Cryosphere, 8, 2063–2074, https://doi.org/10.5194/tc-8-2063-2014, 2014.

Gruber, S. Derivation and analysis of a high-resolution estimate of global permafrost zonation. *The Cryosphere*, *6*(1), 221–233. https://doi.org/10.5194/tc-6-221-2012. 2019.

Gruber, S., Brown, N., Stewart-Jones, E., Karunaratne, K., Riddick, J., Peart, C., Subedi, R., and Kokelj, S. V.: Permafrost Ground Temperature Report: Ground temperature and site characterisation data from the Canadian Shield tundra near Lac de Gras, Northwest Territories, Canada, Northwest Territories Geological Survey, Northwest Territories, Canada, 2019.

Gubler, S., Fiddes, J., Keller, M., and Gruber, S.: Scale-dependent measurement and analysis of ground surface temperature variability in alpine terrain, The Cryosphere, 5, 431–443, https://doi.org/10.5194/tc-5-431-2011, 2011.

Lee, S., Black, T. A., Nyberg, M., Merkens, M., Nesic, Z., Ng, D., and Knox, S. H.: Biophysical Impacts of Historical Disturbances, Restoration Strategies, and Vegetation Types in a Peatland Ecosystem, J Geophys Res Biogeosci, 126, https://doi.org/10.1029/2021JG006532, 2021.

Lee, S.-C., Christen, A., Black, A. T., Johnson, M. S., Jassal, R. S., Ketler, R., Nesic, Z., and Merkens, M.: Annual greenhouse gas budget for a bog ecosystem undergoing restoration by rewetting, Biogeosciences, 14, 2799–2814, https://doi.org/10.5194/bg-14-2799-2017, 2017.

Hernández-Henríquez, M. A., Sharma, A. R., Taylor, M., Thompson, H. D., and Déry, S. J.: The Cariboo Alpine Mesonet: sub-hourly hydrometeorological observations of British Columbia's Cariboo Mountains and surrounding area since 2006, 18, 2018.

MAWN.: Michigan Automated Weather Network. (Retrieved: July 2022 from: https://mawn.geo.msu.edu/)

Morris, J., Hernández-Henríquez, M., and Déry, S.: Cariboo Alpine Mesonet meteorological data, 2017-2021. Zenodo. https://doi.org/10.5281/zenodo.6518969, 2021.

Morse, P. D., Burn, C. R., and Kokelj, S. V.: Influence of snow on near-surface ground temperatures in upland and alluvial environments of the outer Mackenzie Delta, Northwest Territories., Can. J. Earth Sci., 49, 895–913, https://doi.org/10.1139/e2012-012, 2012.

NDAWN.: North Dakota Mesonet Network. (Retrieved: July 2022 from: https://ndawn.ndsu.nodak.edu/).

Obu, J., Westermann, S., Bartsch, A., Berdnikov, N., Christiansen, H. H., Dashtseren, A., et al. Northern Hemisphere permafrost map based on TTOP modelling for 2000–2016 at 1 km2 scale. *Earth-Science Reviews*, *193*, 299–316. https://doi.org/10.1016/j.earscirev.2019.04.023. 2019.

RISMA.: Real-Time In-Situ Soil Monitoring for Agriculture. (Retrieved: July 2022 from: https://agriculture.canada.ca/SoilMonitoringStations/)

RoTimi Ojo, E. and Manaigre, L.: The Manitoba Agriculture Mesonet: Technical Overview, Bulletin of the American Meteorological Society, 102, E1786–E1804, https://doi.org/10.1175/BAMS-D-20-0306.1, 2021.

Spence, C. and Hedstrom, N.: Baker Creek Research Catchment Hydrometeorological and Hydrological Data, 2018a.

Spence, C. and Hedstrom, N.: Hydrometeorological data from Baker Creek Research Watershed, Northwest Territories, Canada, Earth Syst. Sci. Data, 10, 1753–1767, 2018b.

Street, L. E., Mielke, N., and Woodin, S. J.: Phosphorus Availability Determines the Response of Tundra Ecosystem Carbon Stocks to Nitrogen Enrichment, Ecosystems, 21, 1155–1167, https://doi.org/10.1007/s10021-017-0209-x, 2018.

Streletskiy, D. A., Shiklomanov, N. I., & Nelson, F. E. Spatial variability of permafrost active-layer thickness under contemporary and projected climate in Northern Alaska. *Polar Geography*, *35*(2), 95–116. https://doi.org/10.1080/1088937X.2012.680204. 2019.

Tao, J., Koster, R. D., Reichle, R. H., Forman, B. A., Xue, Y., Chen, R. H., and Moghaddam, M.: Permafrost variability over the Northern Hemisphere based on the MERRA-2 reanalysis, The Cryosphere, 13, 2087–2110, https://doi.org/10.5194/tc-13-2087-2019, 2019.

Yukon Geological Survey: Yukon permafrost reports data. In: Yukon Permafrost Database. Government of Yukon, 2021.

---

## Author Response (AR2)

**Summary**

We thank both referees for their helpful comments and we have incorporated their feedback in the revisions. The manuscript has undergone a substantial revision and has incorporated a couple of new sections, as well as some changes to the methodology. These include:

1. Removing GLDAS-CLSM and GLDAS-Noah, as during our revision process, we came to realize that the discontinuities related to GLDAS-Noah and GLDAS-CLSM were related to differences in the driving meteorology between Version 2.0 and Version 2.1 of the products. These have been replaced with FLDAS, which has more a consistent forcing meteorology.

2. Remapping all products to the ERA-Interim resolution (0.75°) instead of the GLDAS resolution (1°).

3. The addition of a new section on soil temperature trends for the individual products (Section 4.4 – Multi-Annual Trends). We have also split our results on the variability of seasonal extremes into a section focused on the individual products (Section 4.5), and a section focused on the ensemble mean (Section 5.3).

4. An exploration of elevation impacts on soil temperature performance, described in the methods (Section 3.3) and Section 4.3 – Spatial Variability of the results.

5. Removing JRA55 from the ensemble mean soil temperature calculation, as its inclusion was found to dramatically increase the bias and RMSE of the ensemble mean.

**Reviewer 1**

P2, L43: Cao et al., 2022 presented improved ERA5-Land soil temperature in permafrost regions using an optimized multi-layer snow scheme. Please consider citing the reference.

We thank the referee for this helpful suggestion and have included Cao et al. (2022) as a reference pertaining to snow thermal insulation in ERA5-Land.

P9, L205: Please use "discontinuous permafrost" rather than "extensive discontinuous permafrost". See Zhang et al., 2000.

Corrected

P21, L353: In latex, $^\circ$C rather than $^\circ$~C

Corrected

P27, L445: Is it possible to show the change time of data assimilation for each product? This could be added to Table 1.

During our revision process, we came to realize that the discontinuities related to GLDAS-Noah and GLDAS-CLSM were related to differences in the driving meteorology between Version 2.0 and Version 2.1 of the products. In the case of CFSR, we diagnosed the changes in soil temperature variability to be mainly related to snow cover issues discussed in Section 4.4 – Multi-Annual trends. Thus the discussion about changes in upper-atmosphere assimilation are no longer relevant to the trends discussion, or to Section 6.2 in the discussion.

**Reviewer 2**

1. The manuscript is too strongly focused on establishing the ensemble mean product. To me, the novel science of the study is to rigorously document and compare the performance of the individual products.
2. In the main paper, I suggest presenting these six numbers for the best-performing product in each category, the most common (e.g. ERA-5) and the overall best-performing products that some users could actually be encouraged to use, and the ensemble mean product.
3. With these numbers, users will have a good basis to decide which product may be most applicable for their application.
4. L. 298: there is a critical difference between "most products" and "all products", so the authors need to add more information on this throughout the entire section. If only most products are worse, then at least one will be similarly good or better, so users can be directed to use this product.
5. For other metrics, single products seem to do better, Fig. 2 for example suggests that ERA-5 with respect to the bias is significantly better than the ensemble mean.

The authors have interpreted these five comments as relating to two main questions:

1. Why did the authors choose to focus the paper on the Ensemble Mean product, rather than a detailed characterization of the performance of individual soil temperature products from reanalysis and land data assimilation system (LDAS)? And

2. Why do the authors not recommend any one individual product as the best choice for estimating soil temperatures in the extratropical northern hemisphere, and the Arctic?

In the following paragraphs, we explain our answers to these two points in detail.

It was clear from previous studies and from our validation of soil temperatures over the extratropical northern hemisphere, that no single reanalysis product provided adequate performance over all regions and times of year (see Figure 2, skill scores). The noticeable decline in performance over the cold season, and over higher latitudes (Figure S1) is present in all products, as evidenced by the substantially lower skill scores relative to the warm season. This made it impractical to recommend a single product and led to the exploration of blending the suite of products together. Blending of multiple observation-based data products is becoming common practice in many subfields of climate science and has been demonstrated to reduce random errors and improve overall performance relative to any individual product (e.g. Mudryk et al. 2015; Dorigo et al., 2017; Gruber et al., 2019; Beck et al., 2019; Cao et al., 2019).  This motivated the development of the ensemble mean product and, therefore, a detailed investigation into the performance of the ensemble mean product was required and became a major focus of the paper.

The primary metric used in the evaluation is the skill score developed by Taylor et al. (2001), which is preferred over singular metrics (such as bias or RMSE), as it incorporates both RMSE as well as aspects of soil temperature variability (normalized standard deviation), similar to the Taylor Diagram. The ensemble mean product showed substantially higher skill than all other products over the cold season (Figure 2), and better performance over higher latitudes (Figure S1); particularly for deeper soil layers. Singular metrics such as bias can be misleading over the annual mean, owing to offsetting errors. For example, the small overall bias of ERA5 and ERA5-Land in Figure 2 are due to offsetting errors over the study region, where soils are too warm over much of the permafrost region, and too cold over more southern regions (See Figure S3); similar to the findings of Cao et al. (2020).

I am missing a clear overview (e.g. a Table) over the key numbers, which for me are: (a) the bias and (b) the RMSE for (1) the entire time period, (2) annual mean values (considering the entire time period), and (3) monthly mean values (considering the entire time period). In the main paper, I suggest presenting these six numbers for the best-performing product in each category, the most common (e.g. ERA-5) and the overall best-performing products that some users could actually be encouraged to use, and the ensemble mean product. Furthermore, for future reference, the authors should provide these numbers for all considered products in the supplement.

Thank you for this suggestion. We have included a table with the annual mean, cold season and warm season bias, RMSE and skill score for each product here (Tables R1 and R2), and in the supplement (Table S1 and S2) and make reference to this table in Section 4.1 (Extratropical Northern Hemisphere Mean) of the revised manuscript. We decided against showing monthly mean values because over such a short time period there would be too much variability between stations to make the averages

meaningful. This is due to the very large range of latitude and continentality between stations and the associated differences in climate and the duration/extent of snow cover.

*Table R1. Summary of the near surface mean bias, RMSE, and skill score for each product for the Extratropical Northern Hemisphere. Metrics are separated into an annual mean metric, and a metric for the cold and warm seasons. The best performing product is listed in bold for each metric, and season. The 95% confidence interval is also included. Multiple bold values may appear if the confidence interval of multiple products overlap for a particular metric and season.*

| Metric | Product | Annual | Cold Season | Warm Season |
|---|---|---|---|---|
| Bias | | | | |
| | CFSR | $-2.63 \pm 0.13°$ C | $-2.37 \pm 0.19°$ C | $-2.8 \pm 0.19°$ C |
| | ERA-Interim | $-2.83 \pm 0.21°$ C | $-4.92 \pm 0.34°$ C | **$-0.69 \pm 0.34°$ C** |
| | ERA5 | $-0.73 \pm 0.14°$ C | **$-0.91 \pm 0.24°$ C** | **$-0.58 \pm 0.24°$ C** |
| | ERA5-Land | **$0.29 \pm 0.14°$ C** | **$0.89 \pm 0.22°$ C** | $-0.35 \pm 0.22°$ C |
| | FLDAS | $-2.62 \pm 0.14°$ C | $-3.42 \pm 0.26°$ C | $-2.18 \pm 0.26°$ C |
| | JRA55 | $-2.15 \pm 0.17°$ C | $4.16 \pm 0.31°$ C | $-7.04 \pm 0.31°$ C |
| | MERRA2 | $-1.55 \pm 0.12°$ C | $-1.88 \pm 0.21°$ C | $-1.33 \pm 0.21°$ C |
| | Ensemble Mean | $-1.68 \pm 0.12°$ C | $-2.1 \pm 0.20°$ C | $-1.32 \pm 0.20°$ C |
| RMSE | | | | |
| | CFSR | $3.89 \pm 0.14°$ C | $3.74 \pm 0.15°$ C | $3.65 \pm 0.15°$ C |
| | ERA-Interim | $4.55 \pm 0.26°$ C | $5.88 \pm 0.33°$ C | **$2.17 \pm 0.33°$ C** |
| | ERA5 | **$2.75 \pm 0.15°$ C** | $3.14 \pm 0.18°$ C | **$2.15 \pm 0.18°$ C** |
| | ERA5-Land | **$2.51 \pm 0.17°$ C** | **$2.67 \pm 0.21°$ C** | **$2.13 \pm 0.21°$ C** |
| | FLDAS | $4.01 \pm 0.13°$ C | $4.60 \pm 0.17°$ C | $3.12 \pm 0.17°$ C |
| | JRA55 | $7.68 \pm 0.17°$ C | $5.87 \pm 0.21°$ C | $8.95 \pm 0.21°$ C |
| | MERRA2 | $2.90 \pm 0.13°$ C | $3.30 \pm 0.15°$ C | **$2.33 \pm 0.15°$ C** |
| | Ensemble Mean | $2.85 \pm 0.13°$ C | $3.17 \pm 0.14°$ C | **$2.31 \pm 0.14°$ C** |
| Pearson Correlation | | | | |
| | CFSR | $0.959 \pm 0.003$ | $0.655 \pm 0.018$ | $0.931 \pm 0.018$ |
| | ERA-Interim | $0.971 \pm 0.002$ | $0.731 \pm 0.014$ | $0.905 \pm 0.014$ |
| | ERA5 | $0.970 \pm 0.003$ | $0.564 \pm 0.021$ | $0.950 \pm 0.021$ |
| | ERA5-Land | $0.970 \pm 0.003$ | $0.644 \pm 0.015$ | $0.954 \pm 0.015$ |
| | FLDAS | $0.970 \pm 0.002$ | $0.734 \pm 0.014$ | $0.939 \pm 0.014$ |
| | JRA55 | $0.552 \pm 0.008$ | $0.570 \pm 0.015$ | $0.402 \pm 0.015$ |
| | MERRA2 | **$0.978 \pm 0.002$** | $0.674 \pm 0.015$ | $0.954 \pm 0.015$ |
| | Ensemble Mean | **$0.982 \pm 0.002$** | **$0.793 \pm 0.011$** | **$0.957 \pm 0.011$** |
| Skill Score | | | | |
| | CFSR | $0.946 \pm 0.007$ | $0.689 \pm 0.019$ | $0.902 \pm 0.019$ |
| | ERA-Interim | $0.896 \pm 0.012$ | $0.62 \pm 0.022$ | $0.955 \pm 0.022$ |
| | ERA5 | $0.951 \pm 0.008$ | $0.685 \pm 0.017$ | $0.955 \pm 0.017$ |
| | ERA5-Land | $0.949 \pm 0.009$ | $0.601 \pm 0.022$ | $0.95 \pm 0.022$ |
| | FLDAS | $0.935 \pm 0.008$ | $0.691 \pm 0.017$ | $0.938 \pm 0.017$ |
| | JRA55 | $0.536 \pm 0.012$ | $0.564 \pm 0.017$ | $0.647 \pm 0.017$ |
| | MERRA2 | $0.954 \pm 0.008$ | $0.73 \pm 0.016$ | **$0.960 \pm 0.016$** |
| | Ensemble Mean | **$0.958 \pm 0.007$** | **$0.805 \pm 0.014$** | **$0.961 \pm 0.014$** |

*Table R2. Summary of the near surface mean bias, RMSE, and skill score for each product for the Extratropical Northern Hemisphere. Metrics are separated into an annual mean metric, and a metric for the cold and warm seasons. The best performing product is listed in bold for each metric, and season. The 95% confidence interval is also included. Multiple bold values may appear if the confidence interval of multiple products overlap for a particular metric and season.*

| Metric | Product | Annual | Cold Season | Warm Season |
|---|---|---|---|---|
| **Bias** | | | | |
| | CFSR | -3.12 ± 0.16° C | -1.63 ± 0.19° C | -4.48 ± 0.19° C |
| | ERA-Interim | -2.71 ± 0.22° C | -2.26 ± 0.29° C | -2.99 ± 0.29° C |
| | ERA5 | -0.54 ± 0.16° C | **0.27 ± 0.20° C** | **-1.34 ± 0.20° C** |
| | ERA5-Land | **0.10 ± 0.17° C** | 1.33 ± 0.21° C | **-1.10 ± 0.21° C** |
| | FLDAS | -3.09 ± 0.19° C | -2.72 ± 0.26° C | -3.71 ± 0.26° C |
| | JRA55 | -1.82 ± 0.19° C | **0.49 ± 0.27° C** | -3.67 ± 0.27° C |
| | MERRA2 | -2.08 ± 0.15° C | -0.70 ± 0.19° C | -3.30 ± 0.19° C |
| | Ensemble Mean | -1.91 ± 0.15° C | -0.95 ± 0.18° C | -2.82 ± 0.18° C |
| **RMSE** | | | | |
| | CFSR | 4.54 ± 0.16° C | 2.98 ± 0.16° C | 5.28 ± 0.16° C |
| | ERA-Interim | 4.05 ± 0.22° C | 3.42 ± 0.27° C | 4.13 ± 0.27° C |
| | ERA5 | **2.73 ± 0.15° C** | **2.10 ± 0.17° C** | **2.98 ± 0.17° C** |
| | ERA5-Land | **2.76 ± 0.17° C** | **2.28 ± 0.20° C** | **2.82 ± 0.20° C** |
| | FLDAS | 4.68 ± 0.15° C | 3.99 ± 0.18° C | 4.83 ± 0.18° C |
| | JRA55 | 4.24 ± 0.16° C | 3.26 ± 0.14° C | 4.86 ± 0.14° C |
| | MERRA2 | 3.44 ± 0.16° C | **2.17 ± 0.15° C** | 4.00 ± 0.15° C |
| | Ensemble Mean | 3.19 ± 0.14° C | **2.20 ± 0.13° C** | 3.69 ± 0.13° C |
| **Pearson Correlation** | | | | |
| | CFSR | 0.817 ± 0.012 | 0.77 ± 0.018 | 0.749 ± 0.018 |
| | ERA-Interim | 0.909 ± 0.007 | 0.857 ± 0.012 | 0.847 ± 0.012 |
| | ERA5 | 0.906 ± 0.01 | 0.849 ± 0.014 | 0.871 ± 0.014 |
| | ERA5-Land | 0.901 ± 0.01 | 0.86 ± 0.011 | **0.881 ± 0.011** |
| | FLDAS | 0.857 ± 0.013 | 0.84 ± 0.014 | 0.833 ± 0.014 |
| | JRA55 | 0.781 ± 0.009 | 0.865 ± 0.011 | 0.753 ± 0.011 |
| | MERRA2 | 0.91 ± 0.008 | 0.856 ± 0.013 | 0.869 ± 0.013 |
| | Ensemble Mean | **0.922 ± 0.007** | **0.892 ± 0.011** | **0.871 ± 0.011** |
| **Skill Score** | | | | |
| | CFSR | 0.792 ± 0.019 | 0.768 ± 0.019 | 0.768 ± 0.019 |
| | ERA-Interim | 0.852 ± 0.019 | 0.751 ± 0.019 | 0.794 ± 0.019 |
| | ERA5 | **0.872 ± 0.017** | 0.825 ± 0.018 | 0.843 ± 0.018 |
| | ERA5-Land | 0.839 ± 0.02 | 0.781 ± 0.022 | **0.864 ± 0.022** |
| | FLDAS | 0.759 ± 0.022 | 0.697 ± 0.02 | 0.782 ± 0.02 |
| | JRA55 | 0.794 ± 0.016 | 0.685 ± 0.018 | 0.762 ± 0.018 |
| | MERRA2 | 0.81 ± 0.021 | 0.814 ± 0.02 | 0.803 ± 0.02 |
| | Ensemble Mean | 0.866 ± 0.017 | **0.853 ± 0.015** | 0.841 ± 0.015 |

In addition, I would like to see a similar quantification for the multiannual trends. With these numbers, users will have a good basis to decide which product may be most applicable for their application. Sect. 4.2/4.3: Please add another section on multi-annual trends, as for the ensemble mean.

The authors thank the reviewer for this suggestion, and have added a new section (Section 4.4 – Multi-Annual Trends) outlining the decadal soil temperature trends for each product. We have included a new figure (Figure 6 in the manuscript) showing various aspects of the annual mean soil temperature trends for each product, alongside station estimates where available, with three supplementary figures displaying maps of the annual mean, DJF, and JJA soil temperature trends (Figure S5 – S7).

Most products generally show annual mean warming (positive soil temperature trends) over North America, with a pocket of regional cooling over Western North America. Over Eurasia, most products, show warming over the annual mean, though CFSR, and the European reanalyses show a region of cooling, particularly over higher latitudes. Similar to skill score, and RMSE, products show greater disagreement over higher latitudes, and during winter.

Fig. 2: It says "bias\RMSE" in the figure, and bias in the caption. I guess this is the bias, not the RMSE?

Yes, this was a typo, and should have read "bias". Figure 2 has been revised to include both bias and RMSE.

Fig. 3: please add (e.g. a histogram with the) number of data points per bin; please add to the caption for which time period (I guess monthly?) the individual values are obtained (same for the following figures 4 and 5).

Yes, the standard deviation is based on the monthly soil temperatures within each station soil temperature bin. We have added this to captions of Figure 3 and the new Figure 4 (standard deviation figure). Figures 3 and 4 now also include a histogram with the number of data points per bin.

Fig. 4: the quality of this figure is very poor, both graphically and content-wise. I cannot really see what to get out of the figure, other than a blob of blue and red dots giving the max/min-range. The authors could consider binning the data as in the previous figures and presenting whisker plots with mean, standard deviation (10/90percentiles) and min/max. At least for the means, it seems to me that this information would be pretty much the same as presented in the previous Fig. 3. The authors should consider if this figure can be moved to the supplement (after drastically improving its quality as suggested above).

We agree that Figure 4 was hard to read and did not really add much new information. This figure has been removed in the revised manuscript.

Fig. 6: to make it easier for the reader, please add a sentence to the caption where a "1:1 match" would be located, i.e. explain the star in the figure better than just "Reference". I also suggest renaming "standard deviation" to "standard deviation of data set".

A product would line up on the 1:1 line if the timeseries at all stations matched perfectly, and have added a sentence to the caption to this effect. We've also renamed "Reference" to "Station" and "standard deviation" to the "standard deviation of the dataset".

(Note this is now Figure 5 in the revised manuscript)

Fig. 8: Please provide more information in the caption, what does "1 model" mean, how is this calculated?

"1 model" refers to the average RMSE and Pearson Correlation of the reanalysis and LDAS individual products themselves (not including the ensemble mean).

(Note: This figure has been removed in the revised manuscript)

Fig. 9: The axis description of Panel B is almost in the Panel A figure; the color axis in Panel B is not specified. This is poor manuscript preparation by the authors.

We have corrected this, and the figure now includes a colorbar in Panel B.

(Note that this figure is now Figure 12 in the revised manuscript)

Sect. 6.1: This is a key question: "why are the reanalysis products so bad and how can they be improved". Despite going in much detail with the validation, it appears that the manuscript cannot add much new insight, pretty much the entire section is about other published studies.

We thank the reviewer for this question. We have used the revision process to reassert the principal contributions of our study, which are articulated in the revised manuscript and summarized here. This study is the first to validate soil temperatures from all major modern reanalysis systems across the extratropical northern hemisphere. This represents a multi-faceted, complex challenge, requiring assembly of a wide variety of reference datasets from sparse observing networks with large differences in data quality and availability. Previous studies generally limited their analysis to a restricted geographical area, or to one product, allowing for more detailed explorations of specific phenomena,

and the drivers of bias in these regions. We also present a comprehensive quantification of product performance across the seasonal cycle, comparing mid-latitude regions and permafrost regions separately. This study is also the first to investigate the value of an ensemble mean (blended) soil temperature product.

Given the extent of the challenge involved in achieving this first set of objectives, a detailed investigation of the primary drivers of bias in each product is beyond the scope of this study. However, our research paves the way for these detailed process studies in future by identifying the sign, magnitude, timing and location of biases.

**Sect. 6.2: How does this impact the calculated trends?**

Discontinuities in reanalysis products typically affect the upper atmosphere (e.g. Hersbach et al., 2020; Shuangguan et al., 2019), though discontinuities have been noted in the deep soil moisture and deep soil temperature values of MERRA2 (Bosilovich et al., 2015), and in high latitude precipitation (Reichle et al., 2017). Most modern reanalysis products employ overlapping spin-up periods to reduce such issues (Hersbach et al., 2020), and during our analysis we were unable to detect any obvious discontinuities (i.e., those emerging above the magnitude of internal variability) in any soil temperature product at the grid cell, regional or hemispheric scale.

We believe that the changes in the variability noted in CFSR in Section 6.2 in the previous version of the manuscript were due in large part to the anomalous soil temperatures in 2009 and 2010 (caused by issues with CFSR snow depth values during these years); discussed in Section 4.4 – Multi-Annual Trends.

**Sect. 6.3: This is a potentially important issue which should be explored some more in the methods section. The authors should at least analyze the altitude of the observation sites with respect to the average altitude at typical scales of reanalysis grid cells, and possibly exclude observations e.g. from "mountain sites" much higher than the average altitude. For those, the comparison is meaningless, as they should (on average) be significantly colder than the reanalysis products, even if these were perfect.**

We thank the reviewer for this suggestion. We investigated the impacts of elevation by separating stations based on their elevation (as obtained from the Copernicus 90m DEM), and grouped stations into three elevation zones (0m – 500m, 500m – 1000m and >1000m). Owing to the small sample size of higher elevation stations (Table R3), we combined the mid and higher elevation stations together in Table R4. While the referee is correct that the RMSE in elevation is substantially larger at higher elevation stations, and that most products underestimate the elevation (Table R3), the mean performance for soil temperature is not substantially different in low- or higher-grid cells with an elevation at or above 500m (Figure R1).

We have added a subsection in the Methods section (Section 3.3) explaining how we assessed the impact of elevation on product performance and a brief discussion of any minor differences in performance over the elevation bands has been added in Section 4.3 – Spatial Variability.

*Table R3. Number of grid cells in each elevation range for the near surface and at depth.*

| Elevation Range | Near Surface Grid Cells | Depth Grid Cells |
|---|---|---|
| *Below 500m* | 310 | 275 |
| *500m – 1000m* | 105 | 87 |
| *1000m +* | 15 | 15 |

*Table R4. Average elevation RMSE (in metres), along with the 95% confidence interval, for each product as a function of station elevation. Biases are calculated for low elevation stations (below 500m), and for stations above 500m.*

| Product | Avg Elevation RMSE (0m - 500m) | Avg Elevation RMSE (500m+) |
|---|---|---|
| *CFSR* | 147.73 ± 24.33 | 589.31 ± 61.38 |
| *ERA5* | 140.64 ± 21.91 | 583.10 ± 61.63 |
| *ERA5-Land* | 155.55 ± 28.76 | 587.35 ± 61.62 |
| *ERA-Interim* | 163.49 ± 17.93 | 580.54 ± 59.14 |
| *FLDAS* | 76.43 ± 14.49 | 156.74 ± 42.53 |
| *JRA55* | 144.00 ± 16.73 | 579.53 ± 59.17 |
| *MERRA2* | 76.05 ± 10.78 | 143.97 ± 32.69 |

[Figure]

*Figure R1. Bias, RMSE and skill score values for grid cells containing stations with an elevation between 0m and 500m (panels A and C) and for elevations 500m+ (panels B and D). The near surface is shown in panels A and B, while the depth is shown in panels C and D.*

**Minor Comments**

L. 22: Please revise this sentence, this not really correct. The first statement with the 800GtC is for permafrost soils only (i.e. only the permafrost, not active layer and permafrost-free areas in the permafrost region), while the second statement refers to the entire soil carbon pool in the permafrost region. The carbon pool in the atmosphere is around 850 GtC, so only the entire soil carbon pool is significantly larger. In addition, the numbers from Tarnocai et al. (2009) are somewhat outdated, better to use the estimates from Hugelius et al. (2014).

We thank the reviewer for highlighting this. The sentence has been modified to read "Roughly 1400 to 1600 gigatonnes of carbon (GtC) is estimated to be stored in soils in permafrost affected regions of the Northern Hemisphere (Hugelius et al., 2014)."

L. 175: please add abbreviation (SS)

Corrected.

Sect. 4.2 Change title to "Seasonal cycle". "Temporal variability" would also include multi-annual temperature developments which is not analyzed here.

This section has been renamed to "Seasonal Cycle".

L. 304: close bracket.
Corrected.

L. 329/332/333: ??

These were referring to Figure 10. The missing figure numbers here have been corrected.

L. 482: make this a proper sentence.

This sentence has been split into two separate sentences as follows: "However, it is also apparent that the uncertainties arising from variations in the number of grid cells included in a station average are substantially smaller than the spread between reanalysis products. During the cold season, the uncertainty in soil temperatures associated with the spread between reanalysis products is often two to three times larger than the uncertainty arising from fluctuations in station availability."

L. 491: I have a hard time finding this in the results section, please specify what exact metric (monthly averages) this refers, and provide a reference to the section or figure where this is presented. For other metrics, single products seem to do better, Fig. 2 for example suggests that ERA-5 with respect to the bias is significantly better than the ensemble mean.

The RMSE of the new ensemble mean is approximately 0.5°C better than the next best product during the cold season over permafrost regions across both depths (Figure S1).

The low overall bias of ERA5 and ERA5-Land arises due to a cancellation of errors over the study region, with generally warm biases over areas underlain by permafrost, and predominantly cold biases over more southern regions – a finding similar to Cao et al. (2020). When the RMSE of these products are considered, it becomes apparent that they still show substantial errors, particularly over the cold season. Conversely at higher latitudes, ERA5 and ERA5-Land show substantial degradations to performance, where the ensemble mean outperforms them (Figure S1).

**References**

Beck, H. E., Pan, M., Roy, T., Weedon, G. P., Pappenberger, F., van Dijk, A. I. J. M., Huffman, G. J., Adler, R. F., and Wood, E. F.: Daily evaluation of 26 precipitation datasets using Stage-IV gauge-radar data for the CONUS, Hydrol. Earth Syst. Sci., 23, 207–224, https://doi.org/10.5194/hess-23-207-2019, 2019.

Bosilovich, M., Akella, S., Coy, L., Cullather, R., Draper, C., Gelaro, R., Kovach, R., Liu, Q., Molod, A., Norris, P. M., Wargan, K., Chao, W., Reichle, R., Takacs, L., Todling, R., Vikhliaev, Y., Bloom, S., Collow, A., Partyka, G. S., Firth, S., Labow, G., Pawson, S., Reale, O., Schubert, S. D., and Suarez, M.: MERRA-2: Initial Evaluation of the Climate, NASA, Maryland, USA, 2015.

Cao, B., Quan, X., Brown, N., Stewart-Jones, E., and Gruber, S.: GlobSim (v1.0): deriving meteorological time series for point locations from multiple global reanalyses, Geosci. Model Dev., 12, 4661–4679, https://doi.org/10.5194/gmd-12-4661-2019, 2019.

Dorigo, W., Wagner, W., Albergel, C., Albrecht, F., Balsamo, G., Brocca, L., Chung, D., Ertl, M., Forkel, M., Gruber, A., Haas, E., Hamer, P. D., Hirschi, M., Ikonen, J., de Jeu, R., Kidd, R., Lahoz, W., Liu, Y. Y., Miralles, D., Mistelbauer, T., Nicolai-Shaw, N., Parinussa, R., Pratola, C., Reimer, C., van der Schalie, R., Seneviratne, S. I., Smolander, T., and Lecomte, P.: ESA CCI Soil Moisture for improved Earth system understanding: State-of-the art and future directions, Remote Sensing of Environment, 203, 185–215, https://doi.org/10.1016/j.rse.2017.07.001, 2017.

Gruber, S., Brown, N., Stewart-Jones, E., Karunaratne, K., Riddick, J., Peart, C., Subedi, R., and Kokelj, S. V.: Permafrost Ground Temperature Report: Ground temperature and site characterisation data from the Canadian Shield tundra near Lac de Gras, Northwest Territories, Canada, Northwest Territories Geological Survey, Northwest Territories, Canada, 2019.

Shangguan, M., Wang, W., and Jin, S.: Variability of temperature and ozone in the upper troposphere and lower stratosphere from multi-satellite observations and reanalysis data, Atmos. Chem. Phys., 19, 6659–6679, https://doi.org/10.5194/acp-19-6659-2019, 2019.

---

## Author Response (AR3)

**Summary**

The authors would like to thank the two anonymous referees for their helpful comments. We have incorporated their feedback in the revisions. In particular, we have shortened Section 3.4 (Regridding of Reanalysis Products and Calculation of Ensemble Mean Soil Temperature), and have substantially reorganized Sections 4.1 (Annual Mean) and Section 4.2 (Seasonal Cycle). As part of this, we have added the annual mean bias and RMSE to Figure 2, and changed the order of Figure 4 and 5 to align with the revised text in Section 4.2. We also made some minor changes to other subsections in Section 4 and Section 5 to improve clarity.

**Referee 1 Comments**

P1, L3-4: The topic of permafrost carbon is of course important, but does it make sense to include it in the abstract since the manuscript does not mention it?

We have removed the mention of permafrost carbon - the sentence now reads:

"Soil temperatures are an important control of many land-atmosphere exchanges and hydrological processes, and permafrost soils are thawing as the climate warms."

P2, L55: Please consider merging this paragraph with the one above since it only contains one sentence

We have merged this paragraph with the above one.

P11, L253: Could you please be more specific about "small to moderate negative"?

By "small to moderate negative cold biases, we are referring to the result that most products biases on the order of roughly 1°C to 4°C in both seasons.

This section has been restructured and we now note the following about biases:
"Most products are biased cold over the annual mean, and display soil temperature biases of between +0.3°C and -3.1°C…"

P12, L280: is the "2)" a typo
This should say "(Figure 2)". We've corrected this.

P13, L324: Consider changing the section title to "soil temperature trend"

We have renamed this section to "Soil Temperature Trends", and have similarly renamed Section 5.2 to "Ensemble Mean Soil Temperature Trends".

P14, L355: Merge this paragraph with the one above since it only contains one sentence
This sentence was removed as the section underwent some reorganizing at the advice of the other referee.

P29, L435: Please double-check, the overall warming trend should be 0.29 ± 0.12. The warming trend of 0.39 ± 0.15 is specifically for continuous permafrost.

We thank the reviewer for pointing this out – we have corrected this.

P30, L474-475: References are not formatted appropriately

Corrected.

P32, L539: figure ?

This figure was removed from the manuscript. The sentence now reads "A robust ensemble mean can be computed with four products **(not shown)**…"

**Referee 2 Comments**

My main concern with this manuscript is the presentation quality, especially the length, structure and level of detail in the text. The text would benefit from a stricter focus on the main message, but is instead riddled with distracting details of the results.

We interpreted the reviewer's comment regarding a stricter message to refer to some reorganization and reformatting of the results section. We found that some of the material in Section 4.1 (Annual Mean) fits better in Section 4.2 (Seasonal Cycle), so we incorporated it into Section 4.2. In addition we added the annual mean bias and RMSE to Figure 2, which is now mentioned explicitly in Section 4.1. As a result, the order of Figures 4 and 5 have changed.

We also made some minor reorganizations to Section 4.3 (Spatial Variability), Section 4.4 (Soil Temperature Trends), Section 4.5 (Variability in Seasonal Extremes), Section 5.1 (Ensemble Mean Validation), Section 5.2 (Ensemble Mean Soil Temperature Trends), and Section 5.3 (Ensemble Mean Variability in Seasonal Extremes) to improve clarity.

There are also many typos and some redundancies of information in the text (I have pointed out some of them below).

We have corrected the typos/redundancies that the reviewer points out below, along with those pointed out by the other reviewer.

Information presented in figures is also on several occasions discussed far after the figure appears, making a confused impression on the reader.

We have attempted to move the relevant figures closer to the sections where they are discussed, where possible.

In all, this demands a lot of any reader and I would urge the authors to improve the structure, length, and language of the manuscript so that this research, when published, will be more easily accessible to the community.

We have worked to shorten and clarify Section 3.4, and many of the subsections in Sections 4 and 5, and have rearranged the figures where possible in LaTex. For example, we have reduced Section 4.1 (Annual Mean) from nearly 500 words, to 275 words, and Section 3.4 from 360 words to 230 words. Section 4.5 has been reduced from approximately 300 to 200 words, while Section 4.5 (Ensemble Mean Variability in Seasonal Extremes) has been reduced from 550 words to 430 words. The authors believe that the manuscript should be easier to follow as a result.

Section 3.4: Much (or most) of this text feels redundant and could possibly be dramatically shortened or merged with previous sections.

We agree that this section contained many redundancies. It has been shortened substantially to improve clarity. In the revised manuscript, we removed the redundant explanations describing how the soil temperatures were calculated, and now refer readers to Table 2 for more information about the soil layers in each product in order to shorten the second paragraph. The third paragraph has been incorporated into a much shorter second paragraph.

Line 2-4: Permafrost soils do not contain twice as much carbon as the atmosphere. This has been corrected later in the text after last reviewer comments. And permafrost does not melt, it thaws

We have removed the mention of permafrost carbon on the suggestion of the other reviewer - the sentence now reads:

"Soil temperatures are an important control of many land-atmosphere exchanges and hydrological processes, and permafrost soils are thawing as the climate warms."

L27-31: Not necessarily relevant/central focus for this work.

We believe that Line 27-29 on the permafrost carbon feedback provide critical motivation as to why it is important to study permafrost soils. That being said, we do realize that Line 30-31 likely fits better with the previous paragraph, so we've moved that sentence up.

L94: "the the"

Corrected.

L104 " term term"

Corrected.

L160 wasn't = was not

Corrected.

L176 Reanalysis/LDAS (but earlier it was stated the only the term reanalysis would be used)

We have changed this to "Reanalysis" here.

L192: σnorm, R, and SS is already defined above.

The redundant explanations have been removed.

L247: is that a good skill score? (A very brief explanation on how to interpret the skill score would be helpful!)

We have added a brief explanation as follows to the text: "The Taylor Skill Score ranges between a minimum of zero and a theoretical maximum of one. A product with a skill score of 1.0 would display a perfect correlation of 1.0 relative to in-situ soil temperatures, and a soil temperature variance identical to that of the in-situ data."

L250: Unclear sentence: Unclear if soil temperatures at depth are too slow in the JRA55 or if they should be slower due to lag of heat transfer to deeper layers.

This section underwent some reorganization, and this particular sentence was changed. However we include the following statement regarding the JRA55 near surface skill score:

"JRA55, however displays an annual mean skill score of 0.54 near the surface, and skill score of 0.79 at depth (Figure 2). This arises because JRA55 uses a simplified land model that uses just a single vertical layer; the soil temperatures used are computed as averages over the soil column that are, therefore, more similar to deeper soil layers than to the surface. Consequently, JRA55 underestimates the seasonal cycle of observed soil temperatures in the near-surface, and the timing of its annual maximum and minimum soil temperatures are offset by roughly a month relative to other products (not shown)."

L366: most or all?

This sentence has been changed to "The ensemble mean soil temperature product shows closer agreement with in-situ soil temperatures than any of the individual product, when all depths, seasons and regions are considered as a whole."

L375: Sentence lacking a period (.)

Corrected.

L376: "The near surface skill of the ensemble mean in the cold season is nearly 10% than the next best product (Figure 2)." 10% what?

Corrected. This should read "The near surface skill of the ensemble mean in the cold season is nearly 10% **higher**.."

---

## Author Response (AR4)

**Summary**

The authors would like to thank the editor for the suggested corrections we had missed in the last revision. The technical corrections below have been corrected, along with some minor changes to the following figures: Figure 3, Figure 4, Figure 5, Figure 9, Figure S1, Figure S2, and Figure S4 for readability in response to the validation check.

**Technical Corrections**

line 105: shift parenthesis around (Smith et al., )

Corrected.

line 106: "have had permafrost records since the 1990s" = have records of permafrost temperatures since...

Corrected.

line 106: shift parenthesis around (CEN)

Corrected.

line 107: same as line 105

Corrected.

line 505: shift parenthesis to Li et al. (2021)

Corrected.

Regarding figures 3, 4(b, c), 9: please ensure that the colour schemes used in your maps and charts allow readers with colour vision deficiencies to correctly interpret your findings. Please check your figures using the Coblis – Color Blindness Simulator (https://www.color-blindness.com/coblis-color-blindness-simulator/) and revise the colour schemes accordingly.

In Figure 3, we changed the line for MERRA2 to be a dashdot style. We also changed the ylim in the 2nd y-axis from 0 – 32000, to 0 – 26000.

[Figure]

In Figure 4, Figure S1 and Figure S2, MERRA2 is now shown as a right-facing triangle.

[Figure]

Figure 4. Taylor Diagram of the cold season and the warm season performance of reanalysis products. Panels A and B refer to the cold season, while panels C and D refer to the warm season. The top panels, (a) and (c) are for the near surface while the bottom panels, (b) and (d) refer to soil temperatures at depth. The concentric rings (solid grey lines) refer to the centralized root mean square error (CRMSE), and a product would have a CRMSE of zero if the timeseries of the reanalysis matched the station data perfectly; with a normalized standard deviation of one, and a correlation of one.

[Figure]

Figure S 1. Taylor Diagram of the cold season and the warm season performance of reanalysis products at depth. (a) and (c) refer to the zone with little to no permafrost, while (b) and (d) refer to the permafrost zone. The left panels, (a) and (b) are for the cold season while the right panels (c) and (d) refer to the warm season. The concentric rings (solid grey lines) refer to the centralized root mean square error (CRMSE), and a product would have a CRMSE of zero if the timeseries of the reanalysis matched the station data perfectly; with a normalized standard deviation of one, and a correlation of one.

[Figure]

*Figure S 2. Taylor Diagram of the near surface cold season and the warm season performance of reanalysis products. (a) and (c) refer to the zone with little to no permafrost, while (b) and (d) refer to the permafrost zone. The left panels, (a) and (b) are for the cold season while the right panels, (c) and (d) refer to the warm season. The concentric rings (solid grey lines) refer to the centralized root mean square error (CRMSE), and a product would have a CRMSE of zero if the timeseries of the reanalysis matched the station data perfectly; with a normalized standard deviation of one, and a correlation of one.*

In Figure 5, we changed the MERRA2 linestyle to be dashdot, and adjusted the ylim for better readability. The ylim is now between 0 and 10, rather than 0 and 13. The ylim on the 2nd yaxis has also been adjusted to between 0 to 26000 to match Figure 3.

[Figure]

*Figure 5. Taylor Diagram of the cold season and the warm season performance of reanalysis products. Panels A and B refer to the cold season, while panels C and D refer to the warm season. The top panels, (a) and (c) are for the near surface while the bottom panels, (b) and (d) refer to soil temperatures at depth. The concentric rings (solid grey lines) refer to the centralized root mean square error (CRMSE), and a product would have a CRMSE of zero if the timeseries of the reanalysis matched the station data perfectly; with a normalized standard deviation of one, and a correlation of one.*

The ylim in Figure 9, Panels A and B has been adjusted to fall between -1.4°C and +1.4°C, and CFSR has been changed to a scatter with a star pattern. A small inset in Panel A is also shown with CFSR only to illustrate the strong 2009 and 2010 anomalies. The linestyle for MERRA2 has also been updated to a dashdot pattern, to match Figure 3 and Figure 5. We also make reference to the inset in Panel A in the text:

*In CFSR (purple), however, the trend is near zero over North America, and tends towards negative in Eurasia, arising because of anomalously cold years in 2009 and 2010 (see inset in Figure 9, Panel A), and anomalously warm periods in the 80s and early 90s at the beginning of the timeseries (Figure 9, Panels A and B).*

[Figure]

*Figure 9. Near surface soil temperature anomalies and trends for each of the reanalysis products. (a) displays the regionally averaged 1982-2018 annual mean soil temperature anomalies for each reanalysis product north of 40ºN over Eurasia, while (b) displays the same, but over North America. (c) exhibits an estimate of the regionally averaged 1985-2010 annual mean decadal*

*soil temperature trend for each of the individual products, and the ensemble mean for comparison (the error bars represent the 95% CI for the mean trend).*

Finally, the linestyle in Figure S4 has been updated to match that of Figure 3, Figure 5 and Figure 9, and the ylim is now between -0.25 and +0.35m to improve readability.

[Figure]

*Figure S 3. DJF Snow depth anomalies for each of the reanalysis products. (a) displays the regionally averaged 1982-2018 DJF snow depth anomalies for each reanalysis product north of 40ºN over Eurasia, while (b) displays the same, but over North America.*